systems biology/biomechanics/neuroscience

cochlear micromechanics, cochlea, hair cell, second filter, critical oscillator

**Author for correspondence:**
Jorge Berger
e-mail: jorge.berger@braude.ac.il

# A flexible anatomical set of mechanical models for the organ of Corti

Jorge Berger[1] and Jacob Rubinstein[2]

[1]Department of Physics and Optical Engineering, Ort Braude College, Karmiel, Israel
[2]Department of Mathematics, Technion, Haifa, Israel

JB, 0000-0002-1900-6707

We build a flexible platform to study the mechanical operation of the organ of Corti (OoC) in the transduction of basilar membrane (BM) vibrations to oscillations of an inner hair cell bundle (IHB). The anatomical components that we consider are the outer hair cells (OHCs), the outer hair cell bundles, Deiters cells, Hensen cells, the IHB and various sections of the reticular lamina. In each of the components we apply Newton's equations of motion. The components are coupled to each other and are further coupled to the endolymph fluid motion in the subtectorial gap. This allows us to obtain the forces acting on the IHB, and thus study its motion as a function of the parameters of the different components. Some of the components include a nonlinear mechanical response. We find that slight bending of the apical ends of the OHCs can have a significant impact on the passage of motion from the BM to the IHB, including critical oscillator behaviour. In particular, our model implies that the components of the OoC could cooperate to enhance frequency selectivity, amplitude compression and signal to noise ratio in the passage from the BM to the IHB. Since the model is modular, it is easy to modify the assumptions and parameters for each component.

# 1. Introduction

Hearing in mammals involves a long chain of transductions [1–7]. Pressure oscillations are collected from the air by the outer ear, and passed by the middle ear to the perilymph in the inner ear, while reducing the impedance mismatch. Across most of the frequency range audible to non-aquatic mammals [8], the wavelength of sound in the perilymph is longer than the entire cochlea; however, the partitioned structure of the cochlea (in which the basilar membrane (BM) responds to the pressure difference between the chambers at each of its sides) gives rise to a travelling surface wave with shrinking wavelength [9,10],

**Figure 1.** Schematic drawing, showing the components of the OoC. TM, tectorial membrane; SM, scala media; IS, inner sulcus; IHB, inner hair cell bundle; OHB, outer hair cell bundle; OHC, outer hair cell; RL, reticular lamina (set of blue segments); HC, Hensen cells; DC, Deiters cell; BM, basilar membrane. The lines that depict the TM and the HC stand for the surfaces where they contact the endolymph in the subtectorial channel. The OHCs will be subdivided further, as shown in figure 2; the top of each OHC will be called 'cuticular plate' (CP). The model for each of these components is spelled out in §3. The star marks the position that is taken as the origin, $x = y = 0$. The right end of the HC is anchored at $(x, y) = (L + L_{HC}, 0)$. For more realistic depictions of the OoC, e.g. [6,15,16]. The slanting direction of the OHBs (exaggerated in this drawing) follows these references; several models assume that in equilibrium the OHBs are perpendicular to the RL [17], or even point slightly to the left [18]. We do not claim that the slanting direction assumed here is valid for every animal. Further motivation for our choice will be raised in §3.6. The motion of the BM, driven by the forces exerted by the tissues above it and by perilymph pressure in the scala tympani under it, will not be studied here.

such that the energy deposited on the partition is concentrated in a short segment of it (less than a millimetre [11]).

The partitioned structure of the cochlea is described, e.g. in [4,6]. By 'partition', we mean the helical strip composed of the BM, the organ of Corti (OoC), which is mounted on the BM, and the tectorial membrane (TM), located immediately above the OoC and separated from it by a thin fluid gap. The BM has highly anisotropic stiffness, mainly due to the presence of radial collagen fibres [12,13]. The stiffness of the BM gradually decreases by two orders of magnitude and its width increases by a factor of about four from the base to the apex of the cochlea; the longitudinal variation of the BM thickness is less conspicuous. Regarding the partition as 'horizontal,' it divides between (i) the scala media (SM), filled with endolymph, and further 'above' it the scala vestibuli, filled with perilymph, and (ii) the scala tympani (ST), which is connected to the scala vestibuli at the apex of the cochlea. The BM acts as an elastic barrier and is exposed to the pressure difference between the SM and the ST, which is large where vibrations are present, whereas the OoC and the TM are shielded from the ST. As such, most of the elastic energy delivered to the cochlear partition resides at the BM.

We will focus on a slice of the OoC, that senses the vibrations at a particular longitudinal position in the BM, transmits them to the corresponding inner hair cell bundle (IHB), and from there to the auditory nerve. From this point of view, the motion of the BM will be the 'input,' and the motion of the IHB will be the 'output.' The shape of the OoC in the basal region of the cochlea is quite different to the shape near the apex; e.g. fig. 20 in [6]; we will have in mind the OoC in the basal region, where higher frequencies are detected, and where the OoC has the greatest impact on low-amplitude amplification and frequency selectivity [14].

Figure 1 is a schematic drawing (not to scale) of a slice of the OoC, showing the components considered in our description. The outer hair cell bundles (OHBs) are attached to the TM, so that when a cuticular plate (the top of an outer hair cell (OHC)) rises, the corresponding OHB tilts in the excitatory direction (clockwise). We note, however, that the IHB is not attached to the TM. We neglect the influence of the inclination of the reticular lamina (RL) on the inclination of the IHB, so that in order to turn the IHB and send a signal to the auditory nerve, endolymph flux in the subtectorial channel is required.

Our aspiration is not necessarily to obtain a precise description of the mechanical parameters of the different components of the OoC, but rather to gain insight into how these components cooperate to achieve its global operation. In particular, we would like to provide possible explanations for the benefits of having IHBs that are not attached to the TM, and of the curious fact that after transforming fluid flow into mechanical vibration, this vibration is transformed back into fluid flow, this time along a narrow channel, involving high dissipation. Other questions we would like to pursue include what is the advantage of having several OHCs, rather than a single stronger OHC, how does an OHC perform mechanical work on the system, and whether there is any role to passive components such as the Hensen cells (HC).

Moreover, we would like to be in a position to investigate broader questions, such as: Could nature have built the OoC differently, or could an artificial OoC be designed in a different way? In particular, we would like to look for possible mechanisms to achieve frequency tuning (output sharply peaked at some frequency for a given input) and amplitude compression (input changes by several orders of magnitude give rise to significantly smaller changes of the output). In §4 and §5, we use our results to suggest plausible answers to most of the questions above.

Many mechanical simulations fall into two very different categories. In some of them the mechanical activity of the OoC is substituted by an equivalent circuit. In other works, the OoC is divided into thousands of pieces, and a finite elements calculation is carried out [16,18–22]. Hybrid methods have also been used (e.g. [23]). Our approach involves postulating a simplified model for each anatomical component of the OoC, with idealized geometry and with as few elements and forces as possible, in order to capture the features that are essential for its functioning. After the models are chosen, Newton's Laws can be meticulously followed. In this way, most of the analysis can be kept analytical, and only a slight numerical treatment is required. Clearly, by following this approach, we depart from reality, but we gain a simple and transparent way of relating between qualitative assumptions in the model and their influence on the OoC behaviour. An appealing feature of our approach is that nonlinearity, including the possibility of bifurcations, can be incorporated naturally from the assumed physiology of the OHC and OHB, without invoking parameters that depend on the sound pressure level (e.g. gain factor [22,23]).

A well-established conclusion is that the OoC compresses the amplitudes and tunes the frequencies of the vibrations transferred from the stapes to the BM. By taking motion of the BM as the input, we will be mainly investigating the more controversial question of whether there could be an alternative or additional filter that provides compression and tuning on the way from the BM to the auditory nerve [24–30]. The conjecture of such a 'second filter' is usually attributed to the motion of the TM, but our analysis indicates that this feature is not necessary.

Models of the OoC abound [17,18,20–22,25,31,32]. We do not intend to compete with existing models or to improve them. Rather, we consider complementary aspects. Our models depart from the bulk of the literature on the subject by considering the cuticular plates (CPs) and the RL as separate bodies; the CPs can form mild bulges or dents in response to the local forces exerted by the endolymph and by the corresponding OHC and OHB. A related attribute is that we take into account the local pressures in the subtectorial channel relative to that in the SM.

# 2. Analytical procedure

## 2.1. Scope and conventions

We deal with a slice of the OoC, so that our analysis is at most two-dimensional. Whenever we mention mass, force, moment of inertia, torque, or flow rate, it should be understood as mass (or force, etc.) per unit thickness of the slice. Our set of models is sufficiently simple to permit analytic integrations over space, and we will be left with a system of differential equations for functions of time, that can be solved numerically. Since these equations are nonlinear, we do not perform a Fourier analysis.

While there are normally three rows of OHCs, some models [17,33] assume a single OHC. However, we found (§4.4) that a second OHC enables us to position the value of the IHB resonance frequency relative to that of the BM. In order to keep the model simple, we do not include a third OHC, although this can be readily done due to the modularity of our platform.

Guided by measurements indicating that, for a given slice of the OoC, the RL pivots as a rigid beam around the pillar cells head [34,35], we take the origin at this pivot point. We will assume that the equilibrium positions of the RL and of the upper border of the HC lie along a straight line, that will be taken as the $x$-axis (that will be envisioned as 'horizontal' and the $y$-axis will point 'upwards').

Traditionally [36], it has been assumed that the relative motion between the TM and the RL, which governs OHC excitation and generates endolymph flow, is predominantly shearing motion. On the other hand, Nowotny & Gummer [37] showed that the subtectorial gap can shrink and expand. Recent measurements (in the apical region) [30,38] showed that for frequencies that are not too far from resonance, the amplitudes of the $x$- and of the $y$-component of this relative motion are of the same order of magnitude. Here we focus on the pulsatile mode [37,39], which is usually disregarded [33,36]. Accordingly, except for rotational and for fluid motion, motion will be restricted to the $y$-direction.

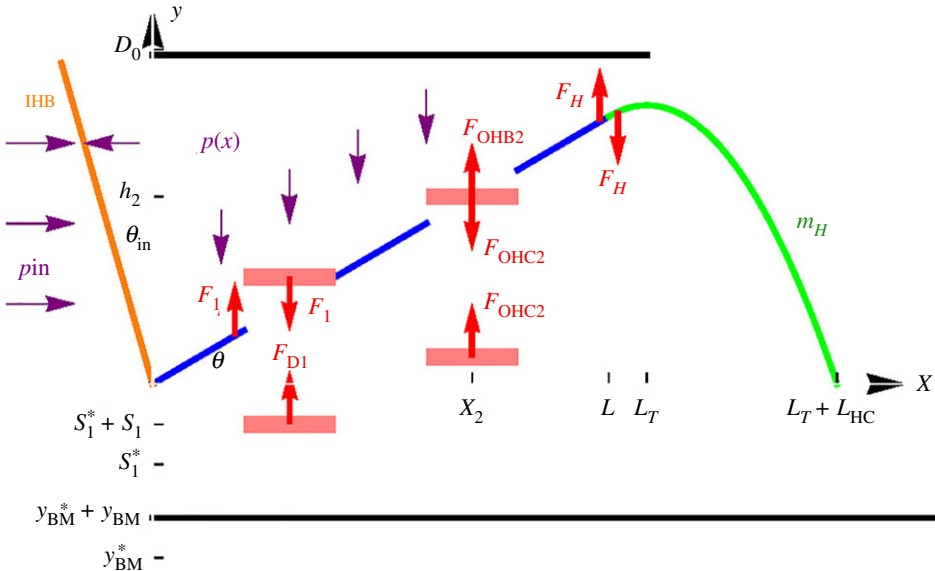

**Figure 2.** Force diagram (not to scale), showing the movable parts in our model and several of the forces that act on them. Each pink rectangle represents a mass $m$. To avoid clutter, analogous quantities that are present in both OHCs are shown in only one of them. The force between a CP and the RL is $F_i = k_{CP}b_i + \beta_{CP}\dot{b}_i$; $F_{D1}$ is shorthand for $k_{D1}(y_{BM} - s_1) - \beta_{D1}\dot{s}_1 + \beta_{D12}(\dot{s}_2 - \dot{s}_1)$; the force exerted by an OHB, $F_{OHBi}$, is given by equation (3.7); the tension of an OHC, $F_{OHCi}$, is given by equations (3.8) and (3.9); the force between the RL and the HC, $F_H$, can be evaluated using equation (3.11). $y_{BM}^*$ and $s_i^*$ are, respectively, the resting heights of the BM and of an OHC-DC interface, and are not required in our equations.

By 'height' of the RL, the HC or the TM, $y_{RL}(x, t)$, $y_{HC}(x, t)$ and $y_T(x, t)$, we imply a position at the surface that is in contact with the endolymph. The thickness of the subtectorial channel is $D(x, t) = y_T(x, t) - y_{RL}(x, t)$ [or $y_T(x, t) - y_{HC}(x, t)$], and we will assume that in equilibrium $D(x, t)$ is constant and denote it by $D_0$. Vertical forces will be considered positive when they act upwards and angular variables will increase in the counterclockwise direction.

## 2.2. Common notations and units

We denote by $L$, $L_{HC}$ and $L_T$ the lengths of the RL, the HC and the TM. The angle of the RL with respect to the $x$-axis is denoted by $\theta$ and $\theta_{in}$ is the angle of the IHB with respect to the $y$-axis. We assume that $|\theta(t)| \ll 1$, so that the projections of the RL and the HC onto the $x$-axis also cover lengths $L$ and $L_{HC}$. Several of the coordinates and forces in our models are illustrated in figure 2.

For an arbitrary function $f(x, t)$ of position and time, we denote $f' := \partial f / \partial x$ and $\dot{f} := \partial f / \partial t$. The absolute value of an arbitrary function $g(t)$ at a given time will be denoted as $|g(t)|$ (with the argument written explicitly), whereas $|g|$ will denote the amplitude of $g$, as defined in appendix A.1.

We have found that the pressure exerted by the endolymph on OoC components can have a major influence on their motion. Since flow of the endolymph is scaled by the height $D_0$ of the subtectorial gap, it is natural to express all quantities in units that involve $D_0$. The unit of length will be $D_0$, the unit of time, $D_0^2/\nu$ and the unit of mass, $\rho D_0^2$, where $\nu$ and $\rho$ are the kinematic viscosity and the density of endolymph. The expected orders of magnitude of these units are $D_0 \sim 10\,\mu\text{m}$, $D_0^2/\nu \sim 10^{-4}\,\text{s}$ and $\rho D_0^2 \sim 10^{-7}\,\text{kg m}^{-1}$. All our variables and parameters are expressed in terms of these units. Using these units might permit scaling results among cochleae of different sizes.

## 3. Detailed modelling

We aim to build a flexible platform in which each anatomical component of the OoC is described by a simple model that translates into a simple differential equation. It is possible to change the model of any of the components by changing just one of the differential equations in the system. In this way, we can readily check how a given feature in the model affects the performance of the entire OoC. Accordingly, the models below may be regarded as initial guesses. Some of them may capture the behaviour of the component that they represent, and others may not.

A *Mathematica* code that integrates our system of differential equations is available as electronic supplementary material and at notebookarchive.org [40]. This code is modular, so that not only the parameters can be varied but also the models.

## 3.1. Subtectorial channel

We denote by $p(x, y, t)$ the pressure in the endolymph and by $v(x, y, t)$ (not to be confused with the kinematic viscosity $v$) the $x$-component of the local velocity. The flow rate in the $x$-direction is

$$Q(x, t) = \int_{y_{RL,HC}(x,t)}^{y_T(x,t)} v(x, y, t)\, dy. \tag{3.1}$$

We assume that the motions of the RL, the HC and the TM are very small in comparison to $D_0$, so that the limits of integration can be set as 0 and $D_0$ (i.e. 1 in our units). We assume that the endolymph is incompressible, so that the net flow entering a region has to be compensated by the expansion of that region and therefore

$$Q' = -\dot{D}. \tag{3.2}$$

Estimating the order of magnitude of the fluid velocity in the subtectorial channel as being similar to that of the fluid in contact with the BM, i.e. the BM velocity, which even for 100 dB SPL does not exceed $10^{-2}$ m s$^{-1}$ [2], leads to a Reynolds number that is at most of the order of $10^{-1}$, so that the flow is certainly laminar. Invoking incompressibility, using the units defined in §2.2, and noting that the term quadratic in velocity is smaller by at least an order of magnitude than the linear terms, the $x$-component of the Navier–Stokes momentum equation reduces to

$$\dot{v} - v'' - \frac{\partial^2 v}{\partial y^2} = -p'. \tag{3.3}$$

By means of a suitable expansion in powers of $D_0/L$ (appendix B) we conclude that the pressure can be taken as independent of $y$ and obtain the approximate relation

$$Q + \frac{\dot{Q}}{10} = -\frac{p'}{12}. \tag{3.4}$$

We assume that the only input is the motion of the BM, whereas the pressure $p(L_T)$ at the exit to the SM is taken as constant. We set $p(L_T) = 0$, i.e. the pressure in the SM will be taken equal to the pressure in the tissues under the RL and the HC.

## 3.2. Reticular lamina

We regard the RL as a straight beam, but exclude the CPs from it, in order to explore the possibility that they bend. The RL obeys the rotational equation of motion

$$I_{RL}\ddot{\theta} = -\kappa_{RL}\theta + \sum F_i x_i + F_H L - \int_{RL} p(x)x\, dx, \tag{3.5}$$

where $I_{RL}$ and $\kappa_{RL}$ are the moment of inertia and the rotational stiffness of the RL, respectively, $F_i$ is the force exerted on the RL by the CP centred at $x = x_i$, $F_H$ is the force exerted on the RL by the HC, and the integration is over the range $0 \leq x \leq L$ excluding the CPs.

## 3.3. Cuticular plates

The CPs are actin rich areas in the apical region of hair cells, where the stereocilia bundles are enrooted. In reptiles and amphibians, the cytoplasma between a CP and the surrounding RL has scarce actin filaments and little mechanical resistance [41–43]. In mammals, the CP has a lip that protrudes beyond the OHC cross-section and extends to adherens junctions with neighbouring cells. The $\beta$-actin density in the CP is much lower than that in stereocilia or in the meshwork through which stereocilia enter the plate, and therefore the CP is expected to be relatively flexible [44]. We will assume that each CP can form a bulge (or indentation) relative to the RL. The length of each CP will be $\ell$ and its

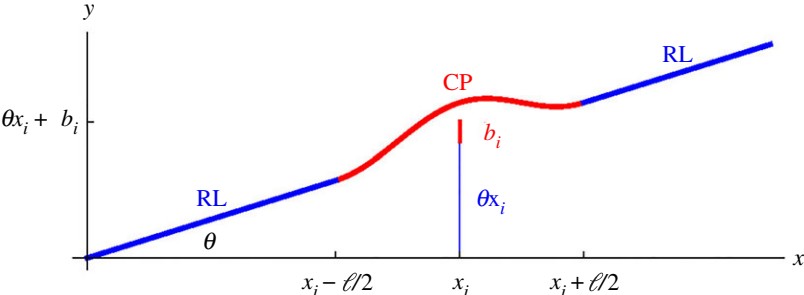

**Figure 3.** Shape of a cuticular plate when it forms a bulge. It extends from $x_i - \ell/2$ to $x_i + \ell/2$ and its average height is $h_i = \theta x_i + b_i$. The scales along the x- and the y-axis are very different.

height $y_i(x) = \theta x + b_i(1 + \cos[2\pi(x - x_i)/\ell])$, where $b_i$ is the average height above the RL, as illustrated in figure 3. Attributing to the CP a mass $m$ and a position $y_i = h_i := \theta x_i + b_i$, its equation of motion is

$$m(\ddot{\theta} x_i + \ddot{b}_i) = -F_i + F_{\text{OHB}i} - F_{\text{OHC}i} - \int_{x_i - \ell/2}^{x_i + \ell/2} p(x)\, dx, \tag{3.6}$$

where $F_{\text{OHB}i}$ is the force exerted by the hair cell bundle and $F_{\text{OHC}i}$ is the tension of the cell. We set $F_i = k_{\text{CP}} b_i + \beta_{\text{CP}} \dot{b}_i$, where $k_{\text{CP}}$ and $\beta_{\text{CP}}$ are restoring and damping coefficients, respectively. The usual assumption that the CPs are fixed within the RL amounts to taking infinite values for $k_{\text{CP}}$ and $\beta_{\text{CP}}$.

## 3.4. Tectorial membrane

The TM is visco-elastic. Its static Young modulus is in the order of tens to hundreds of kPa and has different properties according to the region above which it is located (inner sulcus, RL, or HC) [45]. The poroelastic, electrokinetic, longitudinal-radial coupling and wave properties of the TM are reviewed in [46]. The mechanical properties of the TM are considered to be essential for the tuning ability of the OoC [26,36]. In order to check this assertion, we eliminate the TM motion and replace it by a rigid boundary, located at the constant position $y_T(x) = 1$.

## 3.5. Outer hair cell bundles

We assume that an OHB exerts a force that is a function of its tilt angle, which in turn is a function of $h_i$. We mimic the measured force [47], which has an unstable central region, by means of the expression

$$F_{\text{OHB}i} = \begin{cases} -k_B[h_i - \text{sgn}(h_i)H_i] & |h_i(t)| \geq H_i \\ \frac{k_B H_i \sin(\pi h_i/H_i)}{\pi} & |h_i(t)| < H_i. \end{cases} \tag{3.7}$$

Here $k_B$ defines the stiffness (we will write $k_{\text{Bolt}}$ for Boltzmann's constant), $\text{sgn}(h_i) = h_i/|h_i|$, and $H_i$ is the range of the unstable region. The function $F_{\text{OHB}i}(h_i)$ is shown in figure 4.

Taking $F_{\text{OHB}i}$ as a function of $h_i$ implies that the work performed by the bundle motility vanishes for a complete cycle. Note, however, that if the duration of a cycle is not short compared to the adaptation time [48], $F_{\text{OHB}i}$ becomes history-dependent rather than just a function of $h_i$, and the work that it performs during a cycle does not necessarily vanish.

## 3.6. Outer hair cells

We envision an OHC as a couple of objects, each with mass $m$, connected by a spring. One object is located at the CP and the other at the boundary with the Deiters cell (DC). A special feature of the spring is that its relaxed length can vary. We denote by $c_i$ the contraction of the cell with respect to its resting length, and by $s_i$ the height of the lower object with respect to its average position. We assume that the tension of the OHC has the form

$$F_{\text{OHC}i} = k_C(\theta x_i + b_i - s_i + c_i) + \beta_C(\dot{\theta} x_i + \dot{b}_i - \dot{s}_i), \tag{3.8}$$

with $k_C$ and $\beta_C$ positive constant parameters.

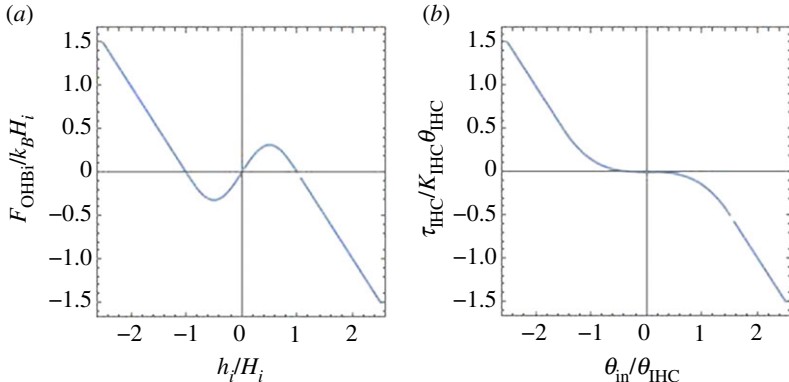

**Figure 4.** (*a*) Restoring force exerted on the CP by OHB *i*, as a function of the height $h_i$ of the CP over its average position, as stipulated in §3.5. (*b*) Restoring torque exerted on the IHB by the inner hair cell, as a function of the bundle deflection $\theta_{in}$, as stipulated in §3.10.

The value of $c_i$ is controlled by the inclination of the hair cell bundle. Guided by [38], we assume that when a CP moves towards the TM the hair bundle bends in the excitatory direction. We assume that $h_i$, scaled by the length $H_i$, acts as a 'degree of excitation,' so that $c_i$ increases with $h_i/H_i$. Since there must be a maximum length, $\Delta$, by which an OHC can contract, the contraction is expected to saturate when the CP has a large deviation from its average position. We take this saturation into account by writing

$$c_i = \Delta \tanh\left(\frac{h_i}{H_i}\right). \tag{3.9}$$

The degree of excitation $h_i/H_i$ may be identified with $Z(X - X_0)/2k_{\text{Bolt}}T$ in equation (3) of [49].

Since $c_i$ is not a function of the distance between the objects on which $F_{\text{OHC}i}$ acts, OHC motility *can* perform non-vanishing work in a complete cycle, as will be spelled out in §4.3.

## 3.7. Deiters cells

We model a DC as a massless spring that connects the lower object in the OHC to the BM (the mass of the DC is already included in $m$). We also include dynamic friction between adjacent lower objects, that encourages oscillation in phase. Denoting by $y_{\text{BM}}$ the height of the BM above its average position, we write

$$m\ddot{s}_i = F_{\text{OHC}i} + k_{\text{D}i}(y_{\text{BM}} - s_i) - \beta_{\text{D}i}\dot{s}_i + \beta_{\text{D}ij}(\dot{s}_j - \dot{s}_i), \tag{3.10}$$

where DC *j* is adjacent to DC *i*. Since DCs are longer for larger *x*, $k_{\text{D}i}$ and $\beta_{\text{D}i}$ can depend on *i*.

## 3.8. Hensen cells

We model the HC as a parabolic strip of evenly distributed mass $m_{\text{H}}$, with its left extreme tangent to the RL and the other extreme pinned at $(x, y) = (L + L_{\text{HC}}, 0)$. These requirements impose $y_{\text{HC}}(x) = \theta[x - (L + L_{\text{HC}})(x - L)^2/L_{\text{HC}}^2]$. The torque exerted on the HC with respect to the pinning point is $F_{\text{H}}L_{\text{HC}} + \int_L^{L+L_{\text{HC}}} p(x)(L + L_{\text{HC}} - x)dx$, and equals the time derivative of the HC angular momentum, $-(m_{\text{H}}/L_{\text{HC}})\int_L^{L+L_{\text{HC}}} \ddot{y}_{\text{HC}}(L + L_{\text{HC}} - x)dx$, leading to

$$F_{\text{H}} = -\frac{m_{\text{H}}}{12}(5L + L_{\text{HC}})\ddot{\theta} - \frac{1}{L_{\text{HC}}}\int_L^{L+L_{\text{HC}}} p(x)(L + L_{\text{HC}} - x)\,dx. \tag{3.11}$$

Since we assume that the pressure vanishes in the SM, we replace the upper limit in the integral with the end of the subtectorial channel. We will take this end over the position where the HC has maximum amplitude, namely, $L_{\text{T}} = L + L_{\text{HC}}^2/2(L + L_{\text{HC}})$.

## 3.9. Inner sulcus

We take the pressure $p_{\text{in}}$ in the inner sulcus (IS) as uniform and proportional to the increase of area (volume per thickness of the considered slice) with respect to the relaxed IS. We write

$$\dot{p}_{\text{in}} = -CQ(0). \tag{3.12}$$

$C$ is some average value of the Young modulus divided by the area (in the $xy$-plane) of the soft tissue that coats the IS and $Q(0)$ is the flow rate for $x = 0$.

## 3.10. Inner bundle

We locate the IHB at $x = 0$ and assume that its length is almost 1, i.e. it almost touches the TM. Models for the torque exerted by the fluid on the IHB abound [19,33,37,50]. We take a simpler approach. The force exerted by viscosity on a segment of the IHB between $y$ and $y + dy$ is proportional to the relative velocity of endolymph with respect to the segment, and we denote it by $\mu[Q(0) + y\dot{\theta}_{in}] \, dy$, where $\mu$ is a drag coefficient and we have replaced $v(y)$ by its average over $y$. On average, the force per unit length is $\mu[Q(0) + \dot{\theta}_{in}/2]$. We identify this force with the pressure difference and write

$$p_{in} - p(0) = \mu\left[Q(0) + \frac{\dot{\theta}_{in}}{2}\right], \tag{3.13}$$

where $p(0)$ is the pressure at $x = 0$.

The torque exerted by viscosity is $-\mu[Q(0)/2 + \dot{\theta}_{in}/3] = 0$. We assume that the moment of inertia of the bundle is negligible and write $\tau_{IHC} - \mu[Q(0)/2 + \dot{\theta}_{in}/3] = 0$, with $\tau_{IHC}$ the torque exerted by the cell. We assume that the inner hair cell does not rotate, and $\tau_{IHC}$ is a function of $\theta_{in}$. It seems reasonable to assume that, in contrast to the OHB, the IHB does not have a central range with negative stiffness, since this could cause sticking of the bundle at any of the angles at which stiffness changes sign. We assume that, as a remnant of the OHB negative stiffness, $\partial\tau_{IHC}/\partial\theta_{in}$ vanishes at $\theta_{in} = 0$ [similarly figure 1(C) in [48]], and write

$$\tau_{IHC} = \begin{cases} -\kappa_{IHC}[\theta_{in} - \text{sgn}(\theta_{in})\theta_{IHC}] & |\theta_{in}(t)| \geq \frac{3\theta_{IHC}}{2} \\ -\frac{4\kappa_{IHC}\theta_{in}^3}{27\theta_{IHC}^2} & |\theta_{in}(t)| < \frac{3\theta_{IHC}}{2}. \end{cases} \tag{3.14}$$

$\tau_{IHC}$ is a smooth function of $\theta_{in}$ and the parameters $\kappa_{IHC}$ and $\theta_{IHC}$ determine its size and the extension of the low stiffness region. $\tau_{IHC}(\theta_{in})$ is shown in figure 4.

We assume that the rate of impulses passed to the auditory nerve is an increasing function of the amplitude $|\theta_{in}|$.

## 3.11. Basilar membrane

We assume that the BM drives the lower ends of the DCs, each of them by the same amount. In the absence of noise, we take $y_{BM} = A\cos\omega_{BM}t$.

## 3.12. Noise

We investigate the ability of the OoC to filter noise present in the input $y_{BM}$; we do not consider noise due to thermal fluctuations in the OoC itself. We mimic white noise by adding to $y_{BM}$ in equation (3.10) four sinusoidal additions $A_N \cos(\omega_j t - \Phi_j)$, where the frequencies $\omega_j$ are randomly taken from a uniform distribution in the range $0 \leq \omega_j \leq 2\omega_{BM}$. $\omega_1$ (respectively $\omega_2, \omega_3, \omega_4$) is re-randomized at periods of time 0.7 (respectively 0.9, 1.1, 1.3). The values of $\Phi_j$ are initially random, and afterwards are taken so that $A_N \cos(\omega_j t - \Phi_j)$ is continuous. $A_N$ is taken so that the average energy added to the DC (for a slice of thickness $D_0$) is of the order of $k_{Bolt}T \sim 4.2 \times 10^{-21}$ J. The initial values of most variables are taken from normal distributions appropriate for average energies of the order of $0.5k_{Bolt}T$ per degree of freedom; these initial values become unimportant after the typical times considered in our results.

## 3.13. Procedure

Equations (3.2) and (3.4) can be integrated analytically over $x$ and, likewise, the integrals of $p$ in equations (3.5), (3.6) and (3.11) are evaluated. After this, using the constitutive relations (3.7), (3.9) and (3.14), we are left with a system of ordinary differential equations for functions of time, that is solved numerically [40].

## 3.14. Parameters

Clearly, parameters vary among species, among individuals, and along the cochlea. We tried to set parameters of reasonable orders of magnitude. The values we took are based on the literature

[6,16,20,31,51–53], when available. When forced to guess, our main guideline was to choose values that lead to large flow for a given amplitude of the input. Additional criteria were fast stabilization, similar amplitudes of $b_1(t)$ and $b_2(t)$, avoidance of beating, resonance frequency in a reasonable range, etc. Some of the parameters have almost no influence.

Since bending of the CPs has not been considered in the literature, the value of $k_{CP}$ deserves explicit discussion. Since the thickness of the CP's lip is roughly a third of its length [44], we expect $k_{CP}$ to be of the order of the lip's Young modulus divided by $3^3$. A range of reasonable values for the RL's Young modulus is stated in [53]. $\beta$-actin and spectrin are relatively scarce in the lip region [44], possibly indicating scarce cross-linking and therefore less resistance to bending; accordingly, we took the Young modulus 50 kPa, close to the lower bound quoted in [53], leading to $k_{CP} \sim 2$ kPa. For $\rho = 10^3$ kg m$^{-3}$, $\nu = 7 \times 10^{-7}$ m$^2$ s$^{-1}$ and $D_0 = 5 \times 10^{-6}$ m, this can be written as $k_{CP} \sim 10^2 \rho \nu^2 / D_0^2$.

The parameters we used in our calculations are listed in table 1.

# 4. Results

## 4.1. Key findings

We regard the maximal contraction of the OHC, $\Delta$, as a control parameter, i.e. the parameter that quantifies the power generated within the system. We find that there is a critical value of the control parameter, $\Delta = \Delta_c$, such that for $\Delta > \Delta_c$ the OoC is unstable and undergoes self-oscillations (non-zero output for zero input), whereas for $\Delta < \Delta_c$ it is stable. If we take $\Delta = \Delta_c$, the OoC becomes a critical oscillator [54–56]. Expressions for the output amplitude close to $\Delta = \Delta_c$ are worked out in appendix C. For the parameters in table 1, we found $\Delta_c = 0.254 D_0$ and in the limit $\Delta \to \Delta_c$ the oscillation frequency is $\omega_c = 5.338$ in units of $\nu/D_0^2$. We stress that these values depend on the parameters we took, and they are valid only for the particular slice being considered. For $\nu = 7 \times 10^{-7}$ m$^2$ s$^{-1}$ and $D_0 = 5 \times 10^{-6}$ m, the critical frequency is $\omega_c/2\pi = 24$ kHz and the critical contraction is $\Delta_c = 1.3$ µm. The length of an OHC is typically $\sim 10 D_0$, so that its critical contraction corresponds to a few percent of its length.

Critical oscillator behaviour can be a great advantage for the purpose of tuning and amplitude compression [55]. Here, we explore the implications of having this behaviour in the 'second filter.' Hence, in the following we study the case $\Delta = \Delta_c$. In this case, if $\omega_c/2\pi$ is near the frequency of the sound wave that is picked by the BM at the considered slice position, then the OoC provides additional tuning, i.e. vibrations of the IHB are more sharply tuned than those of the BM; if $\omega_c/2\pi$ is far from the BM resonance frequency, the OoC provides an alternative mechanism for tuning, i.e. IHB vibrations can be tuned at a frequency at which the BM does not resonate. If $\omega_c/2\pi$ is moderately close to the BM resonance frequency, the OoC can provide moderate additional tuning and shift the resonance frequency of the IHB from that of the BM. In the case that $\Delta$ is below, but moderately close to $\Delta_c$, then the second filter can provide moderate additional tuning and compression. As explained in [55], fluctuations that lead to $\Delta > \Delta_c$ result in spontaneous otoacoustic emissions with frequency $\omega_c/2\pi$.

Figure 5 shows the gain $|\theta_{in}|/|y_{BM}|$ as a function of the frequency, for several amplitudes of $y_{BM}$. Our results show remarkable similarity between the passage from the BM to the IHB and the experimentally known gain of the BM with respect to the stapes [2,57]. In both cases, weaker inputs acquire larger amplification and tighter selectivity. Except for the case of the lowest amplitude, the gain becomes independent of the amplitude far from the resonance frequency. The inset in figure 5 is an expansion of the range $5.1 \leq \omega_{BM} \leq 5.5$. It shows that the gains for moderate amplitudes behave as expected from a critical oscillator in the vicinity of the bifurcation point (see appendix C). As is often the case in critical phenomena, there is also a remarkable similarity between figure 5 and fig. 5b of [32], despite the distinct differences between the considered models.

The gain curves are skewed, providing a faster cut at lower frequencies than at higher frequencies. This feature is complementary to the selectivity provided by the cochlear partition, that provides a fast cutoff for high frequencies.

Indeed, early experiments found that, as the frequency is lowered below resonance, the pressure levels required to excite the auditory nerve or to generate a given electrical response at an inner hair cell grow faster than the pressure levels required to bring about a given vibration amplitude at the BM [58,59]. The credibility of these experiments was limited by the suspicion that the mass or the damage caused by the Mössbauer source or by the reflecting bead used in the measurement of BM vibration could affect its tuning, and also by the large variability [60], which implies that comparison of quantities measured in

**Table 1.** Parameters used in our calculations.

| parameter | $L$ | $L_H$ | $x_1$ | $x_2$ | $\ell$ | $m$ | $m_H$ | $l_{RL}$ | $\kappa_{RL}$ | $k_{CP}$ | $\beta_{CP}$ | $k_c$ | $\beta_c$ | $k_{D1}$ | $k_{D2}$ | $\beta_{D1}$ | $\beta_{D2}$ | $\beta_{D12}$ | $k_B$ | $H_1$ | $H_2$ | $\kappa_{IHC}$ | $\theta_{IHC}$ | $C$ | $\mu$ |
|---|---|---|---|---|---|---|---|---|---|---|---|---|---|---|---|---|---|---|---|---|---|---|---|---|---|
| value | 10 | 10 | 3 | 7 | 2 | 10 | 120 | $2\times10^3$ | $10^3$ | $10^2$ | 3 | 50 | 43 | 400 | 400 | 3 | 3 | 3 | 10 | $6.5\times10^{-3}$ | $5\times10^{-3}$ | 10 | $5\times10^{-3}$ | 2 | 10 |
| definition | 2.2 | 2.2 | 3.3 | 3.3 | 3.3 | 3.3 | 3.8 | 3.2 | 3.2 | 3.3 | 3.3 | 3.6 | 3.6 | 3.7 | 3.7 | 3.7 | 3.7 | 3.7 | 3.5 | 3.5 | 3.5 | 3.10 | 3.10 | 3.9 | 3.10 |

We assume that the maximal contraction of the OHC takes its bifurcation value, which for these parameters is $\Delta_c = 0.254$. The third row indicates the section where the symbol is defined. The system of units is defined in S2.2.

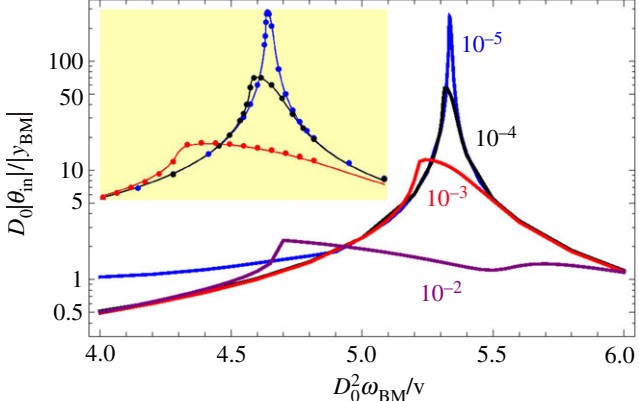

**Figure 5.** Gain supplied by the OoC. $|\theta_{in}|$ is the root mean square (rms) amplitude of the deflection angle of the IHB and $|y_{BM}|$ is the rms amplitude of the height of the BM at the point where it touches the DC, $y_{BM} = A\cos\omega_{BM}t$ ($|y_{BM}| = A/\sqrt{2}$). The value of $A$ is marked next to each curve. In these evaluations, we have ignored thermal noise. Inset: the dots are calculated values for our system and the lines obey equation (C 7) with the fitted values $|B| = 1.8 \times 10^3$, $\alpha = 6.6 \times 10^{-4}$, $\chi_1 = -0.66$ (for the three lines). Our units are specified in §2.2.

different individuals may not be justified. A later experiment [27] compared vibrations at a BM site with the response of auditory nerve fibres innervating neighbouring inner hair cells, and obtained good agreement between BM and nerve responses, provided that BM displacements were high-pass filtered, or BM velocities were considered instead. Still, it could be argued that if for faint amplitudes $|\theta_{in}|$ is very sharply tuned, then the spike of the nerve response curve could infiltrate undetected between consecutive measured points. Also, the variability argument could be reversed to claim that the absence of a second filter in a few cases does not rule out its existence in other individuals or locations.

If the transduction from the BM to the IHB has critical oscillator behaviour, then the amplitude compression at resonance of neural activity should be larger than that of BM motion. Indirect experimental support for this scenario is provided by measurements of the OoC potential [61] and of the ratio between the amplitudes of motion of the RL and the BM [62].

Figures 6 and 7 compare the time dependencies of the input and of the output in the case of a small signal when noise is present. The signal had the form $y_{BM} = A\cos\omega_{BM}t$ during the periods $2000 < t < 4000$ and $6000 < t < 8000$, and was off for $0 < t < 2000$ and $4000 < t < 6000$. We took $A = 3 \times 10^{-5}$ and $\omega_{BM} = 5.329$ (which corresponds to the highest gain for this amplitude). Our model for noise is described in §3.12. The input $y_{total}(t)$ is the sum of the signal and the noise. Panel $a$ in each of these figures shows the entire range $0 < t < 8000$, and the other panels focus on selected ranges.

Figure 6$b$ shows $y_{total}(t)$ in a range such that during the first half only noise is present, whereas during the second half also the signal is on. It is hard to notice that the presence of the signal makes a significant difference. Figure 6$c$ contains three lines: the blue line shows $y_{total}(t)$ during the lapse of time indicated at the abscissa, close to $t = 8000$; the brown line refers to the values of $y_{total}(t)$ at times preceding by $400 \times 2\pi/\omega_{BM} \approx 472$, after the signal had been on for about 1500 time units, and the red line refers to times preceding by $3500 \times 2\pi/\omega_{BM}$, close to the end of the first stage during which the signal was on. Despite the fact that the signal was identical during the three lapses of time considered, there is no obvious correlation between the three lines.

In contrast to figure 6$a$, we see in figure 7$a$ that $\theta_{in}$ is significantly larger when the signal is on than when it is off. The blue, brown and red lines in figure 7$b$ show $\theta_{in}(t)$ for the same periods of time that were considered in figure 6$c$. In this case, the three lines almost coalesce, and are very close to the values of $\theta_{in}(t)$ that are obtained without noise. In particular, we note that the phase of $\theta_{in}(t)$ is locked to the phase of the signal.

Figure 7$c$ shows $\theta_{in}(t)$ for $5995 < t < 6000$, and also for periods of time preceding by 400 and by 3500 times $2\pi/\omega_{BM}$. In the three cases, the signal was off. We can see that the IHB undergoes significant oscillations due to thermal fluctuations even though there is no signal. We also note that there is 'ringing,' i.e. oscillations are larger after the signal was on, and it takes some time until they recover the distribution expected from thermal fluctuations. Unlike the case of figure 7$b$, the phase is not locked, and wanders within a relatively short time. If the brain is able to monitor the phase of $\theta_{in}(t)$, an erratic phase difference between the information coming from each of the ears can be used to discard noise-induced impulses, and a continuous drift in phase difference can be interpreted as motion of the sound source.

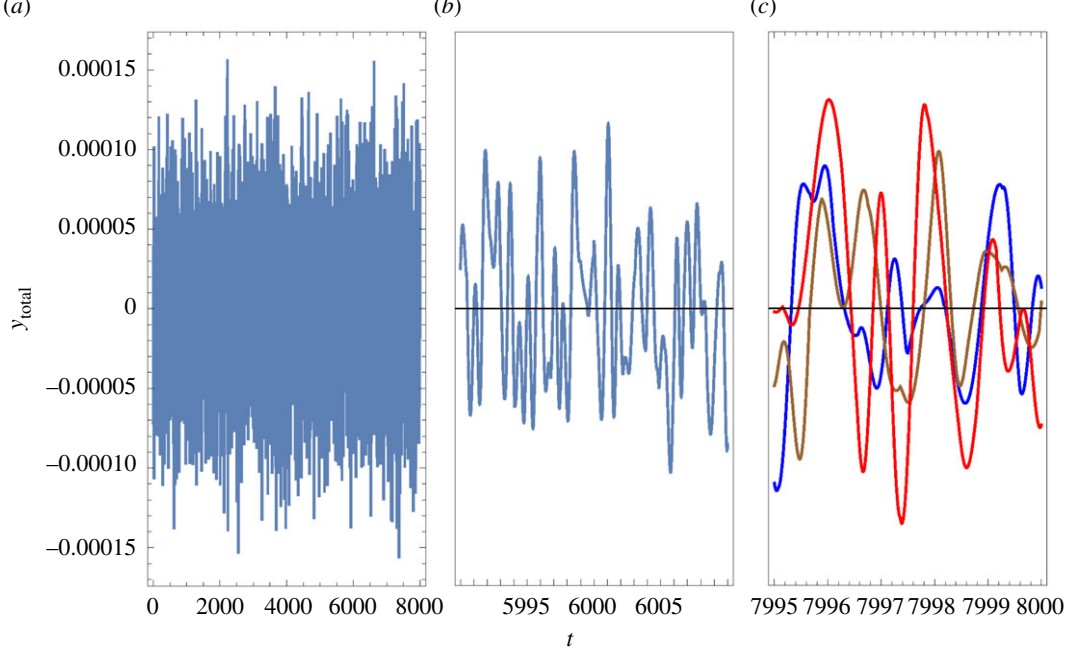

**Figure 6.** Input when noise is present. The height of the BM relative to its equilibrium position is $y_{total}(t) = A \cos \omega_{BM} t + A_N \sum_{j=1}^{4} \cos(\omega_j t - \Phi_j)$, with $A = 3 \times 10^{-5}$, $\omega_{BM} = 5.329$, $A_N = 3.5 \times 10^{-5}$, $\omega_j$ periodically randomized and $\Phi_j$ determined by continuity. (*a*) Entire considered range. (*b*) Range that contains the instant $t = 6000$, at which the signal is switched on. (*c*) Three lines obtained during equivalent periods while the signal was on: the blue line describes the period $7995 < t < 8000$ and the brown (respectively, red) line describes a lapse of time that preceded by 400 (respectively, 3500) times $2\pi/\omega_{BM}$. Our units are specified in §2.2.

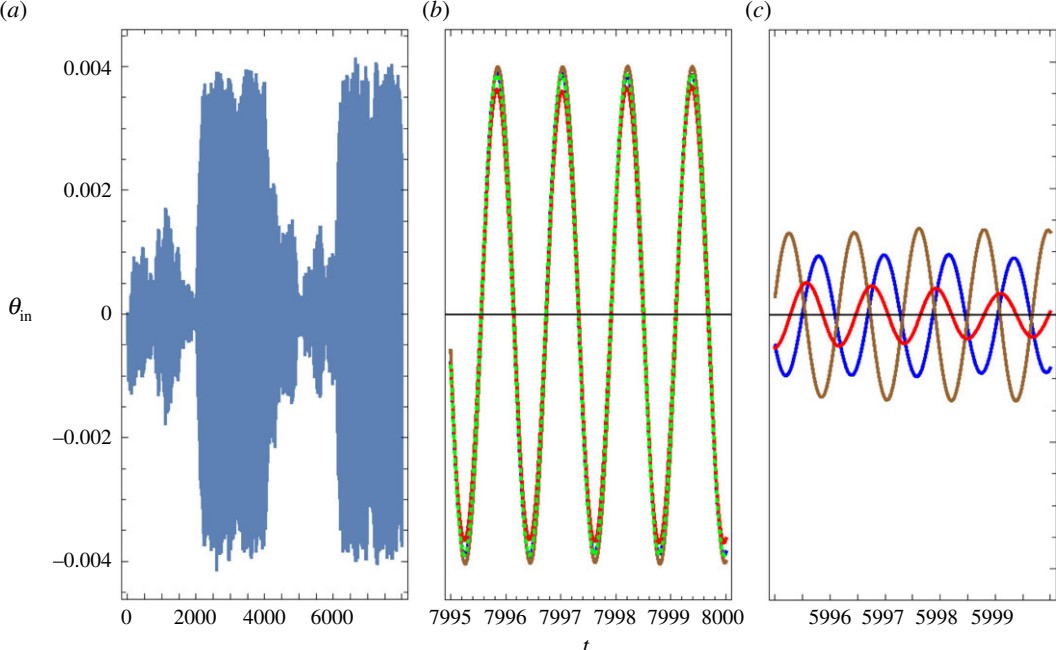

**Figure 7.** Output, $\theta_{in}(t)$, for the situation considered in figure 6. (*a*) Entire range. (*b*) The blue, brown and red lines correspond to the same periods of time shown in figure 6*c*; the dotted green line was obtained by dropping the contribution of noise to $y_{total}(t)$. (*c*) The three time lapses shown in panel *b* have been shifted 2000 units to the left, so that they cover ranges when no signal was present.

Strictly following our models, if the IHB were attached to a fixed point in the TM, it would not move. In a more realistic model, motion of the BM would tilt the pillar cells, leading to inclination of the IHB. Therefore, in the case of an attached IHB, the signal to noise ratio of the IHB's inclination would be

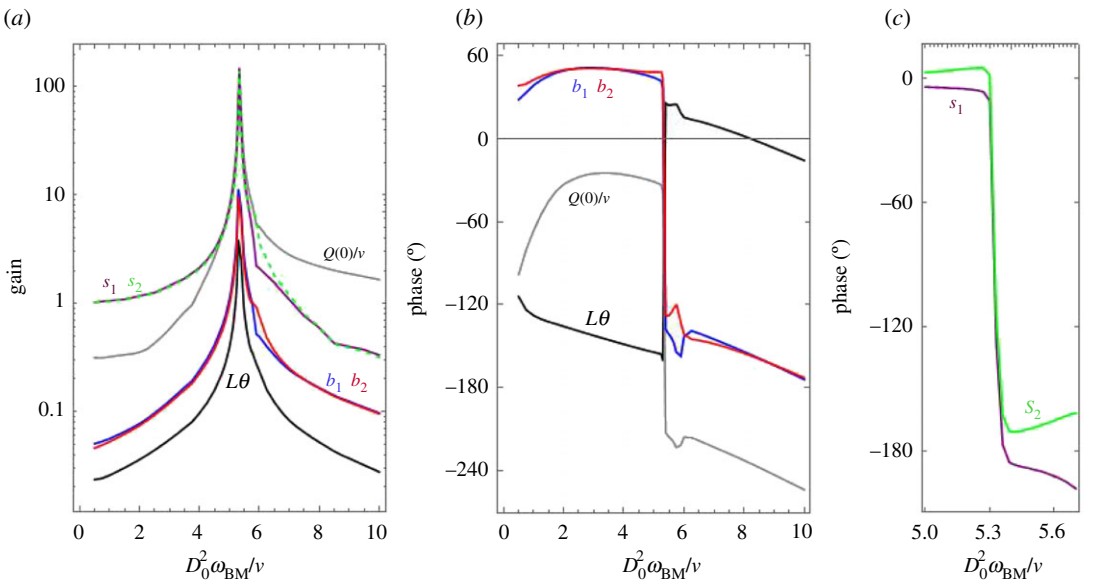

**Figure 8.** Amplitude and phase of several variables, relative to the input $y_{BM} = 10^{-4}D_0\cos\omega_{BM}t$ (which typically corresponds to approx. 50 dB SPL). (*a*) Amplitude, as defined in equation (A 1). For visibility, $s_2$ is depicted by a dashed line. (*b*) Phase by which the variable precedes the input. Phases that differ by an integer number of cycles are taken as equivalent. The phase of a variable is defined as the phase of its first harmonic (see appendix A). (*c*) Phases of $s_1$ and $s_2$ near the resonance. Here and in the following figures noise has been neglected.

similar to that of BM motion. On the other hand, comparison of figures 6 and 7 shows that the signal to noise ratio of $\theta_{in}$ is much larger than that of $y_{BM}$, strongly suggesting one possible answer to the question of why the IHB is not attached to the TM: in this way, the signal to noise ratio increases remarkably.

### 4.2. Motion of each component

Figure 8 shows the amplitudes and phases of $Q(0)/v$, $b_{1,2}$, $s_{1,2}$ and $L\theta$ for a broad range of input frequencies. $b_1$ and $b_2$, and likewise $s_1$ and $s_2$, nearly coincide, except for a small range of frequencies slightly above the resonance, where the motion in the first OHC is considerably smaller than in the second. $L|\theta|$ is roughly three times smaller than $|b_{1,2}|$ and $\theta$ is nearly in anti-phase with $b_{1,2}$ (lags by approx. 200°). The opposite motions of the RL and the CPs may be attributed to incompressibility and to our assumption of a rigid TM, so that when one of them goes up the other has to go down. $Q(0)$ typically lags behind $b_{1,2}$ by ∼80°; following the incompressibility argument, $Q(0)$ is positive when the sum of the subtectorial volumes taken by the CPs, the RL and the HC is decreasing. All the variables undergo a 180° change when crossing the resonance.

At resonance, $|b_{1,2}| \sim 0.5 \times 10^{-3}\ell$, indicating that the CPs are just moderately bent.

We note that close to the resonance the amplitudes of $s_{1,2}$ are larger than those of $b_{1,2}$ and $L\theta$. This result is in agreement with the finding of a 'hotspot' located around the interface between the OHCs and the DCs, where vibrations are larger than those of the BM or of the RL [63].

Separate motion of the CPs and the RL has not been detected experimentally. We could argue that the lateral spatial resolution of the measuring technique did not distinguish between the CPs and the surrounding RL, so that the measured motion corresponds to some average, but the spot size reported in [34] (less than a µm) excludes this possibility. In the case of [34], there was electrical simulation, and no input from the BM. The most likely possibility is that the TM recedes when the CPs go up, so that the RL does not have to recede and is mainly pulled by the CPs. For a relevant comparison with experiment, the RL motion in figure 8 would have to be interpreted as motion relative to the TM, which was within the limits of reproducibility in [34].

A marked difference between [28] and figure 8 is the absence of phase inversion when crossing the resonance, possibly indicating that the maximum gain (amplitude of RL motion divided by the amplitude of BM motion) occurs at a frequency beyond the range considered in fig. 5 of [28] (which includes the maximum of BM motion). A sharp decrease of the phase of the RL relative to the BM occurs in [62].

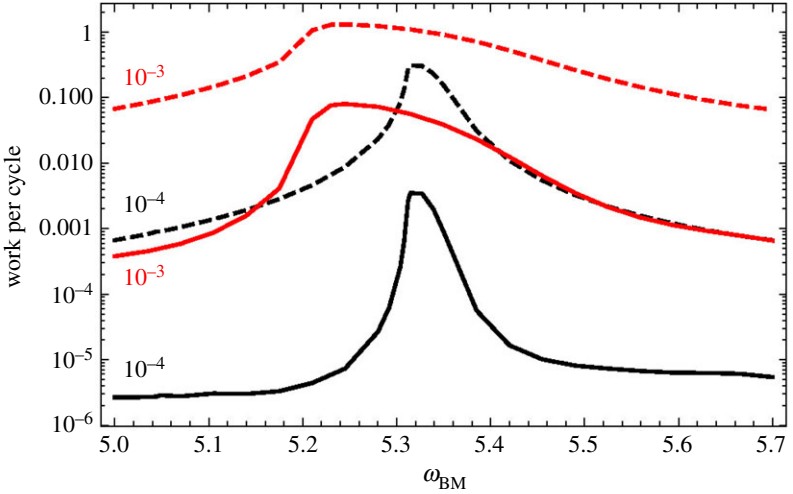

**Figure 9.** Work performed during a cycle for frequencies close to resonance. The dashed lines refer to the work delivered by electromotility, $W_{OHC}$, and the continuous lines to the work taken from the BM, $-W_{DC1} - W_{DC2}$. $y_{BM} = AD_0 \cos\omega_{BM}t$ and the value of $A$ is shown next to each line.

## 4.3. Mechanical energy transfer

The power delivered by the electromotility of OHC $i$ is $-k_C c_i(\dot{h}_i - \dot{s}_i)$. Using equation (3.9) and dropping the terms that give no contribution through a complete cycle, the work performed by electromotility during a complete cycle is

$$W_{OHC} = k_C \Delta \sum_{i=1}^{2} \int \tanh\left(\frac{h_i}{H_i}\right) \dot{s}_i \, dt, \tag{4.1}$$

where integration involves a complete cycle. Since both $h_i$ and $s_i$ undergo a phase inversion when crossing the resonance, the sign of $W_{OHC}$ remains unchanged.

Similarly, the work per cycle performed by DC $i$ on the BM is

$$W_{DC_i} = -A k_{Di} \omega_{BM} \int s_i \sin\omega_{BM}t \, dt. \tag{4.2}$$

$W_{DCi} > 0$ if and only if the phase of $s_i$ is in the range between $0°$ and $180°$ (or equivalent, i.e. larger than $2n \times 180°$ and smaller than $(2n + 1) \times 180°$ for some integer $n$). We see from figure 8c that very near the resonance $W_{DC1}$ and $W_{DC2}$ are both negative, indicating that the OoC takes mechanical energy from the BM. For $\omega_{BM} < 5.30$ (but still in the range shown in this figure), $W_{DC1} < 0$, $W_{DC2} > 0$, and the opposite situation occurs for $\omega_{BM} > 5.37$.

Figure 9 shows the values of these works close to the resonance frequencies, for $A = 10^{-4}$ and $A = 10^{-3}$. Most of the energy required for motion in the OoC is supplied by electromotility, and a small fraction is taken from the BM.

## 4.4. Amplification of the travelling wave

So far we considered the effect of the OHC's electromotility on the motion of the IHB. However, as mentioned in the previous subsection, the OoC may also perform work, denoted $W_{DC}$, on the BM itself. Although in this paper we take the BM motion as input, it is instructive to analyse the dependence of $W_{DC}$ upon different parameters.

We recall that the accepted explanation for active tuning by the cochlea is the amplification of each Fourier component of the travelling wave along the segment between the oval window and the place where this component resonates [6,64], followed by attenuation beyond this place. In our model, $W_{DC1} + W_{DC2}$ is the only exchange of mechanical energy between the considered slice of the OoC and its surroundings; the larger this work, the larger the amplification of the wave. Within a more realistic model, the energy exchange described here should be regarded as a contribution to the energy exchange between the BM and the OoC. In the case of figure 9, energy is taken from the travelling wave, leading to attenuation.

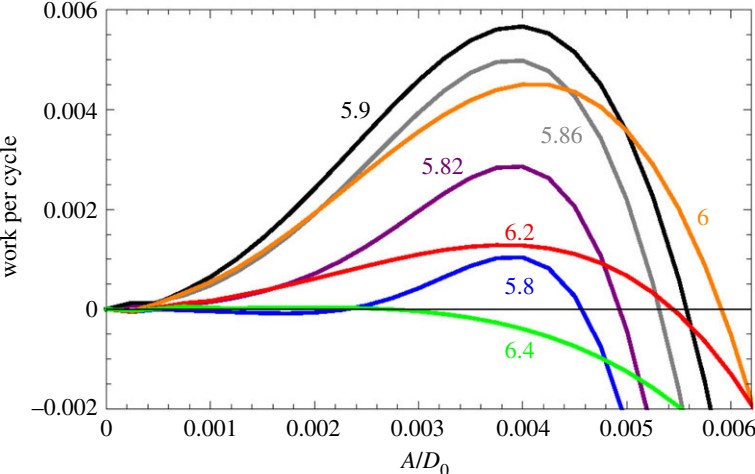

**Figure 10.** Work performed on the BM as a function of the amplitude of the BM oscillations. The parameter $\omega_{BM}$ is shown next to each curve. The travelling wave is amplified if this work is positive and attenuated if it is negative. After many cycles, the amplitude of the BM oscillations would be largest for $\omega_{BM} \approx 6$.

With the parameters of table 1, amplification would occur for $5.8 \lesssim \omega_{BM} \lesssim 6.4$, as shown in figure 10. The work performed on the BM depends on the amplitude of $y_{BM}$ and can even change sign. If this work is positive/negative the amplitude will increase/decrease, thus approaching the amplitude at which $W_{DC1} + W_{DC2} = 0$.

Contrary to the accepted explanation, the amplification range in figure 10 lies above $\omega_c$. This could be the case if the resonance of the 'first filter' lies above that of the second, but the situation can also change if the parameters are slightly varied. For example, if we raise $k_{D2}$ by 10%, to 440, $\Delta_c$ becomes 0.290, $\omega_c$ becomes 5.506, and the travelling wave is amplified in the range $4.9 \lesssim \omega_{BM} \lesssim 5.4$, as shown in figure 11. Conceivably, the advantage of having several OHCs per slice (rather than a single stronger OHC) is the possibility of adjusting the resonance frequencies of both filters, so that they cooperate rather than interfere with each other.

Figures 10 and 11 indicate that the travelling wave is attenuated for frequencies below the considered ranges. However, we should note that the energy transferred for given work per cycle is not proportional to the travelled distance, but rather to the travelling time. Therefore, the largest influence will be that of the slices where the travelling wave is slow, close to the resonance of the first filter. The number of cycles that the travelling wave is expected to undergo while passing through a given region is estimated in appendix D. Dependence of amplification on time rather than on distance could help explain the unexpected results obtained in [64].

## 4.5. Time dependence of the output

Figure 12 shows $\theta_{in}(t)$ for $A = 10^{-5}$ and frequencies near resonance. The blue envelope was obtained at resonance frequency, $\omega_R = 5.334$, the pink envelope at $\omega_{BM} = 5.34$ and the green envelope at $\omega_{BM} = 5.32$. In the case of resonance, the output amplitude raises monotonically until a terminal value is attained. Out of resonance, the amplitude starts increasing at the same pace as at resonance, overshoots its final value, and then oscillates until the final regime is established.

This initial behaviour implies that the IHB starts reacting to the input if $\omega_{BM}$ is moderately close to $\omega_R$, before it can tell the difference between these two frequencies. Conversely, for a given $\omega_{BM}$, there are several slices of the OoC with a range of frequencies $\omega_R$ close to $\omega_{BM}$ that start reacting to this input. As an effect, all these slices send a fast alarm signalling that something is happening, before it becomes possible to discern the precise input frequency.

In contrast with a forced damped harmonic oscillator, when out of resonance, motion of the OoC does not assume the frequency of the input even after a long time, but is rather the superposition of two modes, one with the input frequency $\omega_{BM}$, and the other with the resonance frequency $\omega_R$. If $\omega_{BM} = (n_1/n_2)\omega_R$, where $n_{1,2}$ are mutually prime integers, then the motion has period $2n_2\pi/\omega_R$. Figure 13 shows $\theta_{in}(t)$ for $\omega_{BM} = (2/3)\omega_R$ and for $\omega_{BM} = (4/3)\omega_R$.

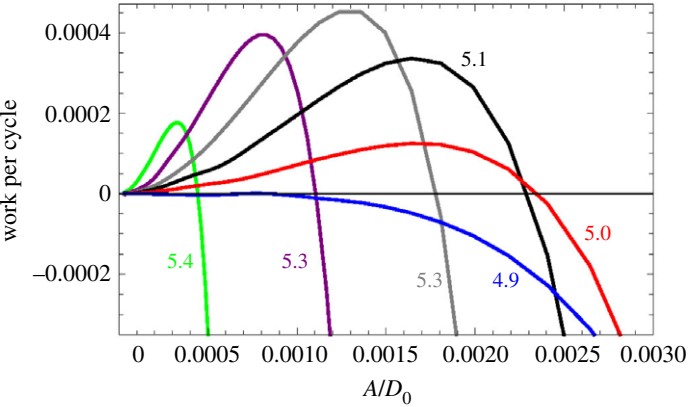

**Figure 11.** Similar to figure 10, this time for $k_{D2} = 440$ rather than 400.

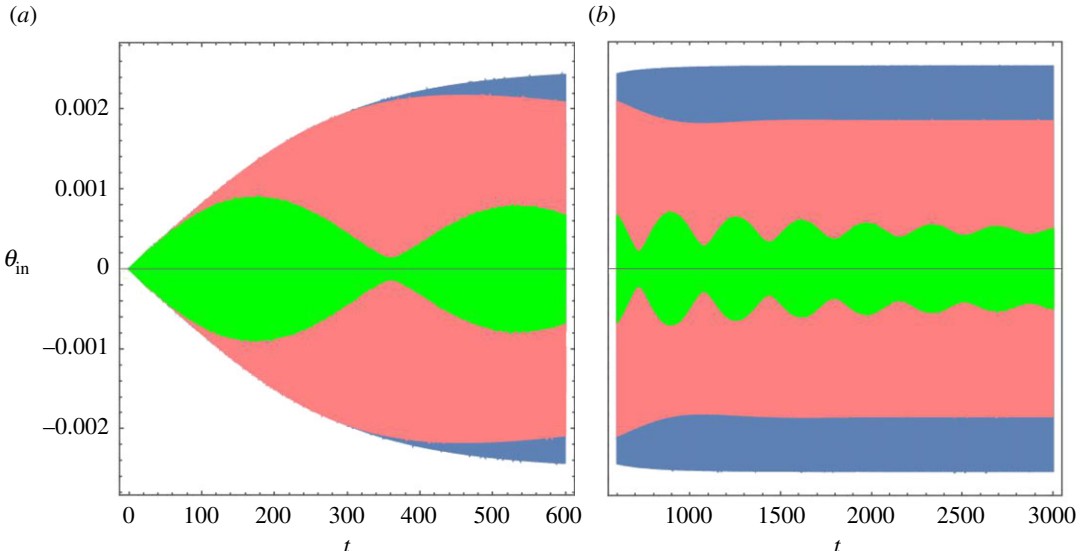

**Figure 12.** Angle of the IHB as a function of time in response to $y_{BM} = 10^{-5} D_0 \cos \omega_{BM} t$. Blue: resonance frequency, $\omega_{BM} = \omega_R = 5.334$; pink: $\omega_{BM} = 5.34$; green: $\omega_{BM} = 5.32$. (a) $0 \leq t \leq 600$. (b) $t \geq 600$. $\omega_R$ approaches the critical frequency $\omega_c$ in the limit of small amplitude.

### 4.6. Nonlinearity

Since the constitutive relations (3.7), (3.9) and (3.14) are nonlinear, it is not a surprise that a sinusoidal input can result in a non-sinusoidal output, but rather contain higher harmonics. For example, if the peak to peak amplitude of the BM vibration is $0.002 D_0$ and its frequency is $\omega_R / 2\pi$, then the vibration of the IHB (after the periodic regime is established) is not sinusoidal, but is rather approximately proportional to $\cos \omega_R t + 0.034 \cos [3(\omega_R t + 0.70)] + 0.005 \cos [5(\omega_R t - 0.34)]$. Expanding $\theta_{in}(t)$ in a Fourier series, $\theta_{in}(t) = \sum_{n=0}^{\infty} a_n \cos[n(\omega_{BM} t + \phi_n)]$, we studied the amplitude dependence of the coefficients at the resonant frequencies. We obtained that the even harmonics vanish. Taking the origin of time such that $\phi_1 = 0$, we found the values reported in table 2.

## 5. Discussion

We have built a flexible framework that enables testing many possibilities for the mechanical behaviour of the components of the OoC. The models we used imply that even by taking the basilar membrane motion as an input, the OoC can behave as a critical oscillator, thus providing a second filter that could enhance frequency selectivity and improve the signal to noise ratio. This framework can be used to explore and theoretically predict different effects that would be hard to observe

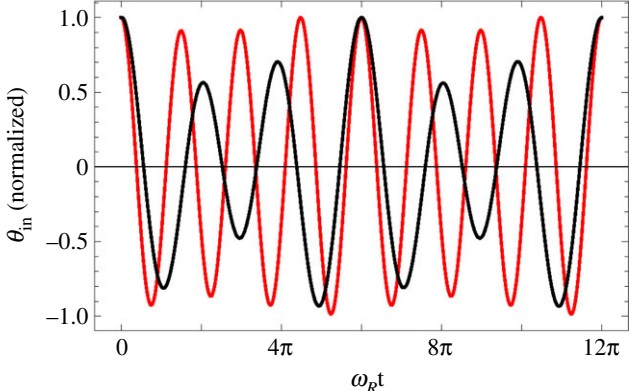

**Figure 13.** $\theta_{in}(t)$ during a short period of time. Black: $\omega_{BM} = (2/3)\omega_R$; red: $\omega_{BM} = (4/3)\omega_R$. $t$ is the time elapsed after a maximum of $\theta_{in}$, roughly 4000 time units after the input was turned on. $A = 10^{-5}$.

**Table 2.** $\theta_{in} \approx a_1\cos\omega_{BM}t + a_3\cos[3(\omega_{BM}t + \phi_3)] + a_5\cos[5(\omega_{BM}t + \phi_5)]$.

| $A$ | $a_1$ | $a_3/a_1$ | $\phi_3$ | $a_5/a_1$ | $\phi_5$ |
|---|---|---|---|---|---|
| $10^{-3}$ | $1.25 \times 10^{-2}$ | 0.0344 | 0.70 | 0.0050 | −0.34 |
| $10^{-4}$ | $5.75 \times 10^{-3}$ | 0.0097 | 0.67 | 0.0001 | −0.51 |
| $10^{-5}$ | $2.54 \times 10^{-3}$ | 0.0017 | 0.66 | 0.0000 | |

$A$ is the peak value of the input and $\omega_{BM}$ equals the resonance frequency. $\phi_{3,5}$ are the phases with respect to the first harmonic of $\theta_{in}$.

experimentally. Although the models considered here are oversimplifications, they enabled us to obtain features that are compellingly akin to those observed in the real OoC.

According to our models, the fluid flow at the IHB region is driven by the vertical motion of the CPs, the RL and the HC. We point out that other mechanisms are also possible [37,39]; for instance, the flow could be due to shear between the TM and the RL, due to squeezing of the IS, or due to deviation of part of the RL from the $x$-axis, implying an $x$-component of its velocity when it rotates.

For comparison of the relative importance of each of these mechanisms, we examine the peak values that we obtained for $A = 10^{-4}$ at resonance frequency. For $Q(0)$, which in our units equals the average of $v(y)$ over $y$, we found $\sim 2 \times 10^{-2}$. The vertical velocity of the CPs is less than $10^{-2}$. From here we expect that the shear velocity of the RL with respect to the TM will be less than that, and the average fluid velocity even smaller.

The peak value of $\dot{\theta}$ is $\sim 5 \times 10^{-4}$. Assuming that the length of the RL that invades the IS is $\sim 4D_0$, squeezing would cause a flux rate of $\sim 10^{-3}$. It therefore seems that the mechanism that we have considered is the most important, providing a sort of self-consistency check. In the case of a flexible TM, $\theta$ would be larger and the flux due to squeezing would grow accordingly.

The following sections describe examples of possible modifications of our models. Some of them we have already studied and others have not been studied thus far.

## 5.1. Bundle motility

Bundle motility can be eliminated from the model by setting $H_i = 0$ in equation (3.7) (but not in (3.9)). We still obtain that the OoC can behave as a critical oscillator, but the critical value for OHC contraction rises to $\Delta_c = 0.262$. Our conclusion is thus that bundle motility helps attainment of critical oscillator behaviour, but is not essential.

## 5.2. Removal of the HC

This was done by setting $L_T = L$ and $F_H = 0$. The bifurcation value of $\Delta$ increased to $\Delta_c = 0.273$, suggesting that an advantage of the HC is reduction of the amount of contraction required to achieve criticality. An

intuitive explanation could be that due to their lower stiffness (neglected in our models) the HC undergo comparatively large deformations and can pump fluid into and out of the subtectorial space. The comparison may be somewhat biased by the fact that our parameters were optimized with the HC included.

## 5.3. Natural extensions

In order to describe a situation as it occurs in nature, our models should consider flexibility of the TM. A model with TM that just recedes would be easy to implement, but a realistic model that includes shearing should also allow for motion of the base of the IHB.

Our models could deal with the longitudinal dimension along the cochlea ($z$) by taking an array of slices, with parameters and input $y_{BM}$ that are functions of $z$. The interaction between neighbouring slices could be mechanical, mediated by the phalangeal processes and deformation of the TM, or hydrodynamic, mediated by flow along the IS and the SM.

For simplicity, in equation (3.9), $c_i$ is an odd function of $h_i$. In reality, OHCs contract by a greater amount when depolarized than what they elongate when hyperpolarized. Equation (3.9) corresponds to the assumption that there are equal probabilities for open and for closed channels [49]. We have found that the asymmetry between contraction and elongation is essential for demodulation of the envelope of a signal, as it occurs in [61].

Instead of adding elements to the set of models, an interesting question is how much can be taken away while still maintaining critical oscillator behaviour. We can show that a system of two particles, with a 'spring' force between them of the form (3.8) that depends on the position of one of the particles, and with appropriate restoring and damping coefficients, behaves as a critical oscillator with an unusual bifurcation diagram. The critical control parameter of this 'bare' oscillator (with the same parameters used in table 1) is considerably smaller than the value of $\Delta_c$ that we found for the OoC. These bare oscillators (one for each OHC) drive the entire OoC.

Data accessibility. All code used in calculating results and generating the presented figures is available as electronic supplementary material and at https://www.notebookarchive.org/models-for-organ-of-corti–2020-01-2aqmevw/ [40].

Authors' contributions. J.R. suggested the problem and formalized appendix B. J.B. performed the numerical analysis and wrote the initial draft. Both authors were active in developing the model and critically revising and approving the final version.

Funding. This research was supported by grant no. 890/16 from the Israel Science Foundation.

Acknowledgements. We are indebted to Anders Fridberger, David Furness, Karl Grosh, James Hudspeth, Daibhid Maoiléidigh, Yehoash Raphael and Luis Robles for their answers to our inquiries.

# Appendix A. Periodic non-sinusoidal functions

## A.1. Amplitude

The amplitude of a periodic, or approximately periodic, function $f$ will be defined as the root mean square deviation from its average,

$$|f| := \left( \int_{t_1}^{t_2} f^2(t) \, \mathrm{d}t/(t_2 - t_1) - \left[ \int_{t_1}^{t_2} f(t) \, \mathrm{d}t/(t_2 - t_1) \right]^2 \right)^{1/2}, \tag{A 1}$$

where $t_2 - t_1$ is an integer number of periods.

## A.2. Phase differences

We consider two real functions, $f_1(t)$ and $f_2(t)$, that have the same period $2\pi/\omega$. We define the 'phase' $\phi$ of $f_2$ with respect to $f_1$ by the value that maximizes the overlap between these functions when the time is advanced in $f_1$ by $\phi/\omega$, i.e. by the value that maximizes $\oint f_1(t + \phi/\omega) f_2(t) \, \mathrm{d}t$.

Equivalently, if we write $f_i(t) = \sum_{n=0}^{\infty} a_{ni} \cos[n(\omega t + \phi_{ni})]$, we have to maximize $\sum_{n=1}^{\infty} a_{n1} a_{n2} \cos[n(\phi + \phi_{n1} - \phi_{n2})]$, implying $\sum_{n=1}^{\infty} a_{n1} a_{n2} n \sin[n(\phi + \phi_{n1} - \phi_{n2})] = 0$. We note that a dc component in any of the functions has no influence on the phase. If $f_1(t)$ and $f_2(t)$ have the same shape, then $\phi_{n1} - \phi_{n2}$ is independent of $n$ and $\phi = \phi_{12} - \phi_{11}$.

In the case of quasi-sinusoidal functions, such that $|a_{n1}a_{n2}/a_{11}a_{12}| < \epsilon \ll 1$ for $n > 1$, we look for a solution $\phi = \phi_{12} - \phi_{11} + O(\varepsilon)$. We expand $\sin[n(\phi + \phi_{n1} - \phi_{n2})] = \sin[n(\phi_{12} - \phi_{11} + \phi_{n1} - \phi_{n2})] + n\cos[n(\phi_{12} - \phi_{11} + \phi_{n1} - \phi_{n2})](\phi - \phi_{12} + \phi_{11}) + O(\epsilon^2)$ and obtain

$$
\phi = \phi_{12} - \phi_{11}
$$
$$
- \frac{\sum_{n=2}^{\infty} a_{n1}a_{n2}n \sin[n(\phi_{12} - \phi_{11} + \phi_{n1} - \phi_{n2})]}{a_{11}a_{12} + \sum_{n=2}^{\infty} a_{n1}a_{n2}n^2 \cos[n(\phi_{12} - \phi_{11} + \phi_{n1} - \phi_{n2})]} \tag{A 2}
$$
$$
+ O(\epsilon^2).
$$

In this article, $f_1(t)$ is proportional to $\cos \omega t$, so that the phase depends solely on the first harmonic of $f_2(t)$ and becomes

$$
\phi = \phi_{12} = \arctan2\left[\oint \sin \omega t \, f_2(t) \, \mathrm{d}t, \oint \cos \omega t \, f_2(t) \, \mathrm{d}t\right]. \tag{A 3}
$$

We note that the phase is not additive, i.e. the phase of $f_3$ with respect to $f_1$ does not necessarily equal the phase of $f_2$ with respect to $f_1$ plus the phase of $f_3$ with respect to $f_2$.

# Appendix B. Fluid flow in a narrow channel with small rapid wall motion

The channel is defined by $T = \{(x, y)| \ 0 < x < L, \ \xi(x, t) < y < D_0\}$. The flow problem is characterized by three non-dimensional parameters

$$
\varepsilon = D_0/L, \quad \zeta = D_0^2\omega/\nu, \quad \xi/D_0 = O(\delta), \tag{B 1}
$$

where $2\pi/\omega$ is the oscillation period (in time) of $\xi$, and $\nu \sim 1 \ \mathrm{mm}^2 \, \mathrm{s}^{-1}$ is the kinematic viscosity. Typical values for the length parameters above are

$$
D_0 \sim 5 \, \mu\mathrm{m}, \quad L \sim 50 \, \mu\mathrm{m}, \quad \xi \sim 5 \, \mathrm{nm}.
$$

Thus, $\varepsilon \sim 0.1$, while $\delta \sim 10^{-3}$. We shall work under the canonical scaling $\zeta = \alpha\varepsilon$, where $\alpha = O(1)$.

The fluid velocity $(v, u)$ and pressure $p$ satisfy the time-dependent Stokes equation

$$
\nu\Delta v = \frac{1}{\rho}\frac{\partial p}{\partial x} + \frac{\partial v}{\partial t}, \tag{B 2}
$$
$$
\nu\Delta u = \frac{1}{\rho}\frac{\partial p}{\partial y} + \frac{\partial u}{\partial t} \tag{B 3}
$$

and

$$
\frac{\partial v}{\partial x} + \frac{\partial u}{\partial y} = 0. \tag{B 4}
$$

Here $\Delta$ is the Laplacian operator. No-slip boundary conditions are assumed on the channel's lateral boundary.

To convert the problem to a non-dimensional formulation, we scale $(v, u)$ by $\bar{P}D_0^2/(\nu\rho L)$, where $\bar{P} = \rho\nu\omega\delta/\varepsilon^2$ is the scale for $p$. We further scale $x$ by $L$, $y$ by $D_0$, and time by $1/\omega$. Finally, we introduce the scaling $\xi_t = \delta D_0\omega\eta_t$, where $\eta(x, t)$ is dimensionless and the subscript denotes derivative. Substituting all of this into the fluid equations, and retaining the original notation for the scaled variables, we obtain

$$
\varepsilon^2 v_{xx} + v_{yy} = p_x + \alpha\varepsilon v_t, \tag{B 5}
$$
$$
\varepsilon^2 u_{xx} + u_{yy} = \varepsilon^{-1}p_y + \alpha\varepsilon u_t \tag{B 6}
$$
$$
\text{and} \quad v_x + \varepsilon^{-1}u_y = 0. \tag{B 7}
$$

## B.1. First-order expansion

We expand $v = v^0 + \varepsilon v^1 + \cdots$ and similarly for $p$, $u$, and the flux $Q = \int_0^1 v(x, y) \, \mathrm{d}y$. To leading order $p^0 = p^0(x, t)$, and $u^0 = u^0(x, t)$ due to (B 6) and (B 7). However, the no-slip boundary conditions imply $u^0 = 0$. To leading order in $\delta$ the horizontal motion of the wall is negligible up to $\varepsilon^3$, and we retain only the vertical motion. Therefore, the kinematic boundary condition at $y = 0$ is

$$
u(x, y = 0, t) = \varepsilon\eta_t. \tag{B 8}
$$

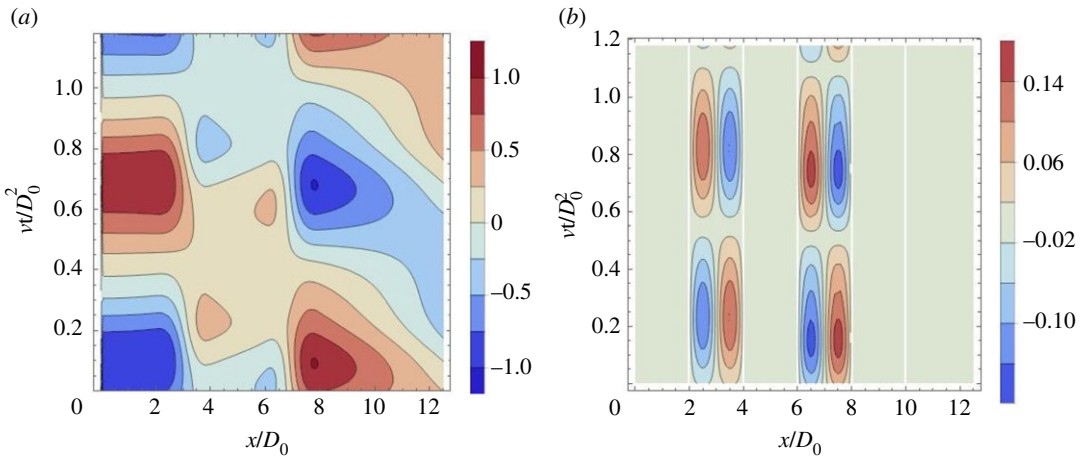

**Figure 14.** (a) Contour plot of normalized pressure gradient, $(D_0^3/\rho v^2)\partial p/\partial x$, as a function of position and time, obtained using (B 13) and thus neglecting $\varepsilon^2 v_{xx}$ in (B 5). (b) y-average of the neglected term, $(D_0^2 \bar{v}) \int_0^{D_0} dy \partial^2 v/\partial x^2$. The white lines are places where $v_{xx}$ is discontinuous. The time span describes one cycle, beginning and ending when $\theta$ assumes its most negative value. For disambiguation, all quantities in the legends and in this caption are dimensional. The colour scale bars are different for each graph.

The leading order term $v^0$ satisfies $v_{yy}^0 = p_x^0$ with boundary conditions $v^0(x, 0, t) = v^0(x, 1, t) = 0$. Therefore,

$$v^0(x, y, t) = \frac{p_x^0}{2}(y^2 - y), \quad Q^0 = -\frac{p_x^0}{12}. \tag{B 9}$$

Integrating the incompressibility equation (B 7) over (0, 1), and since to leading order $u = \varepsilon u^1$, we obtain

$$Q_x^0 = -\int_0^1 u_y^1 dy = \eta_t. \tag{B 10}$$

Combining equations (B 9) and (B 10) provides an equation for the pressure $p_{xx}^0 = -12\eta_t$. Given the boundary motion $\eta(x, t)$, this equation, together with boundary conditions for $p^0$, can be solved to find the pressure and from it the velocity $v^0$ and the flux $Q^0$.

### B.2. Second-order expansion

Since $u^0 = 0$, it follows from equation (B 6) that also $p^1$ satisfies $p^1 = p^1(x, t)$. At the next order, we obtain

$$v_{yy}^1 = p_x^1(x, t) + \alpha v_t^0(x, y, t), \quad v^1(x, 0, t) = v^1(x, 1, t) = 0. \tag{B 11}$$

Using equation (B 9), $v^0$ can be expressed in the alternative form $v^0(x, y, t) = -6\, Q^0(x, t)(y^2 - y)$. Solving equation (B 11) for $v^1$, we find

$$v^1 = \frac{p_x^1}{2}(y^2 - y) - \alpha\frac{Q_t^0}{2}(y^4 - 2y^3 + y).$$

Integrating $v^1$ over (0, 1), we obtain

$$Q^1 = -\frac{p_x^1}{12} - \frac{\alpha Q_t^0}{10}. \tag{B 12}$$

Addition of (B 9) and (B 12) gives the following equation, exact up to $O(\varepsilon)$:

$$Q + \zeta\frac{Q_t}{10} = -\frac{p_x}{12}, \tag{B 13}$$

which is equivalent to equation (3.4).

Similarly, up to $O(\varepsilon)$, $v(x, y, t) = -6Q(x, t)(y^2 - y) - \zeta Q_t(x, t)(5y^4 - 10y^3 + 6y^2 - y)/10$. We recall that $Q(x, t)$ is available from the solution of the system of differential equations in our code. Once $v(x, y, t)$ is known, $u$ can be obtained from (B 4) and the boundary conditions, and the full equations (B 2) and (B 3) can be checked for self consistency. We have found that while the expansion above was carried out for values of $\zeta$ smaller than 1, numerical evidence indicates that equation (B 13) is valid for much

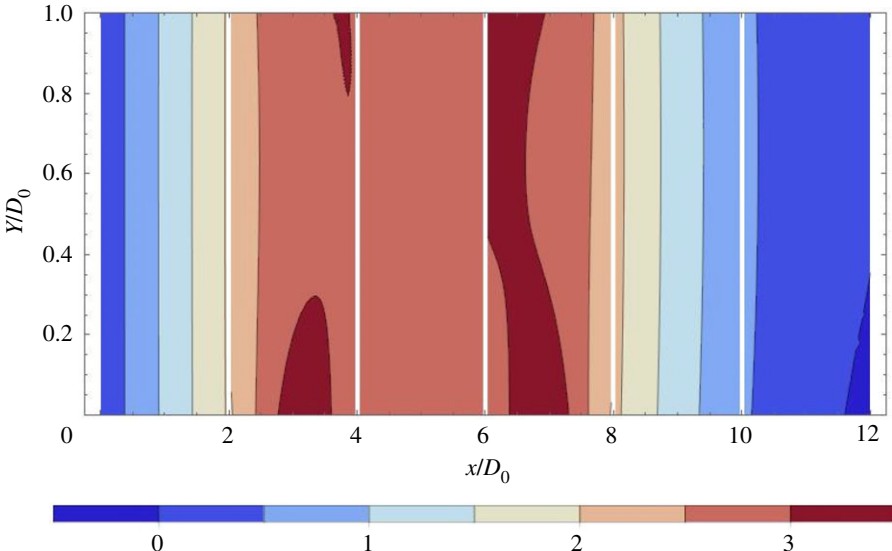

**Figure 15.** Pressure $p(x, y, t = 1.2\pi/\omega_{BM})$ in the subtectorial channel. The pressure unit in the colour scale bar is $\rho v^2/D_0^2$. At the moment depicted in this snapshot, the RL is moving downwards and the CPs are moving upwards. At the white lines, the pressure is discontinuous, but since the $y$-dependence is small the discontinuity is not visible in the figure. For $t \neq 1.2\pi/\omega_{BM}$, $|p(x, y = 0.5D_0, t) - p(x, y = 0, t)|$ is typically smaller.

larger values of $\zeta$. For instance, we consider a representative problem with $\zeta \sim 5$. Expansion up to $O(\varepsilon)$ entirely drops $v_{xx}$ when evaluating $p_x$ in (B 5). Support for this approximation can be based on figure 14, where we see that $v_{xx}$ is significantly smaller than $p_x$. Similarly, figure 15 shows that $p$ is essentially independent of $y$.

# Appendix C. Critical oscillators

Let us deal with an oscillator in which the signal $Y$ can be expressed in terms of the response $X$ in the form

$$Y = A(\omega, \Delta)X + B|X|^2 X + o(|X|^3), \tag{C 1}$$

such that $A(\omega_c, \Delta_c) = 0$. $(\omega_c, \Delta_c)$ is called a 'bifurcation point.' Let us write $\Omega = \omega - \omega_c$, $\delta = \Delta - \Delta_c$ and assume that $B$ can be approximated as constant and $A$ can be expanded as

$$A = B(\alpha\, e^{i\chi_1}\Omega + \beta\, e^{i\chi_2}\delta), \tag{C 2}$$

with $\alpha, \beta > 0$ and $\chi_{1,2} \in \mathbb{R}$.

In order to have a spontaneous response without any signal, $\alpha\, e^{i\chi_1}\Omega + \beta e^{i\chi_2}\delta + |X|^2$ has to vanish. In this case, from the imaginary part we obtain

$$\Omega(\delta) = -\frac{\beta \sin\chi_2}{\alpha \sin\chi_1}\delta, \tag{C 3}$$

and then, from the real part,

$$|X|^2 = -\frac{\beta \sin(\chi_1 - \chi_2)}{\sin\chi_1}\delta. \tag{C 4}$$

Equation (C 4) indicates that non-vanishing spontaneous responses occur either for $\delta > 0$ or for $\delta < 0$, depending on whether the signs of $\sin(\chi_1 - \chi_2)$ and $\sin\chi_1$ are opposite or the same. In our case, $\Delta$ is the maximal contraction of the OHCs and spontaneous responses were found for $\delta > 0$.

Let us now consider forced oscillations, $Y \neq 0$. From (C 1) and (C 2), we have

$$|Y|^2/|X|^2 = |B|^2[\alpha^2\Omega^2 + \beta^2\delta^2 + 2\alpha\beta\cos(\chi_1 - \chi_2)\Omega\delta \\ + 2(\alpha\cos\chi_1\,\Omega + \beta\cos\chi_2\,\delta)|X|^2 + |X|^4]. \tag{C 5}$$

In particular, for $\Delta = \Delta_c$,

$$|Y|^2/|X|^2 = |B|^2[\alpha^2 \Omega^2 + 2\alpha \cos \chi_1 \, \Omega |X|^2 + |X|^4]. \tag{C 6}$$

In our case, the signal is the deviation of the BM from its equilibrium position, the response is the inclination of the IHB, and (C 6) predicts the gain

$$\frac{|\theta_{\text{in}}|}{|y_{\text{BM}}|} = \frac{1}{|B|\sqrt{\alpha^2(\omega - \omega_c)^2 + 2\alpha \cos \chi_1 \, (\omega - \omega_c)|\theta_{\text{in}}|^2 + |\theta_{\text{in}}|^4}}, \tag{C 7}$$

where $B$, $\alpha$ and $\chi_1$ do not depend on $\omega$ or $|y_{\text{BM}}|$.

For small amplitudes and close to the bifurcation point, and for appropriately fitted values of $\Delta_c$, $\omega_c$, $|B|$, $\alpha$, $\beta$, $\chi_1$ and $\chi_2$, our results are in good agreement with equations (C 3), (C 4) and (C 7).

# Appendix D. Number of cycles during which the travelling wave is amplified/attenuated

We want to estimate the number of cycles $n_{\text{cy}}$ experienced by a wave of frequency $\omega_{\text{BM}}$ as it travels across the region $z_1 \leq z \leq z_0$, where $z_0$ is the position (distance from the oval window) of the slice we consider and $z_1$ is the position where the wave starts to be amplified or attenuated significantly.

The dispersion relation can be obtained from equations (2.17) and (2.40) (neglects damping) in [6]:

$$k\tanh(kh) = \frac{\omega_{\text{BM}}^2}{a[1 - \omega_{\text{BM}}^2/\omega_0^2(z)]}, \tag{D 1}$$

where $k$ is the wavenumber, $h$ the height of the chamber above or below the partition, $a$ is a constant and $\omega_0(z)$ is the first-filter resonant frequency at position $z$.

For $kh \ll 1$ and $\omega_{\text{BM}} \ll \omega_0(z)$, (D1) becomes $k^2 h = \omega_{\text{BM}}^2/a$, and therefore $a = V^2(0)/h$, where $V(0)$ is the speed of the travelling wave in the long wavelength limit. For $\omega_{\text{BM}}$ close to $\omega_0(z)$, $kh$ is significantly larger than 1 and (D1) becomes

$$k = \frac{h\omega_{\text{BM}}^2\omega_0^2(z)}{V^2(0)[\omega_0^2(z) - \omega_{\text{BM}}^2]}. \tag{D 2}$$

The number of cycles is $n_{\text{cy}} = (2\pi)^{-1} \int_{z_1}^{z_0} k(z) \, dz$. Assuming that $d\omega_0/dz = -\lambda\omega_0$ with constant $\lambda$, and using (D2) we obtain

$$\begin{aligned} n_{\text{cy}} &= \frac{h\omega_{\text{BM}}^2}{2\pi\lambda V^2(0)} \int_{\omega_0(z_0)}^{\omega_0(z_1)} \frac{\omega_0 d\omega_0}{\omega_0^2 - \omega_{\text{BM}}^2} \\ &= \frac{h\omega_{\text{BM}}^2}{4\pi\lambda V^2(0)} \ln \frac{\omega_0(z_1)^2 - \omega_{\text{BM}}^2}{\omega_0(z_0)^2 - \omega_{\text{BM}}^2}. \end{aligned} \tag{D 3}$$

Taking $h = 0.0005$ m, $\omega_{\text{BM}} = 2\pi \times 5$ kHz, $\lambda = 150$ m$^{-1}$ [11] and $V(0) = 15$ m s$^{-1}$ [65], we obtain $h\omega_{\text{BM}}^2/4\pi\lambda V^2(0) \approx 1$.

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
