## [Peer Review File · Royal Society Open Science]

Review History

RSOS-200675.R0 (Original submission)

Review form: Reviewer 1

Is the manuscript scientifically sound in its present form?

No

Are the interpretations and conclusions justified by the results?

No

Is the language acceptable?

No

Do you have any ethical concerns with this paper?

No

Have you any concerns about statistical analyses in this paper?

No

Recommendation?

Reject

Comments to the Author(s)

This study aims to develop a simplified analytical model to study the OoC behaviour. The text needs to be reviewed by a person fluent in English. In addition, the text needs to be revised structurally. For example, some terms used in the text is not scientific (e.g., hoping to find... instead of making a hypothesis), referencing is not adequate at all, the abstract looks like part of the introduction and does not convey the summary of the work adequately. The introduction section is a blend of abstract, methods and conclusions. The introduction section does not state the current knowledge and most up-to-date findings that should be used as a base for most of the assumptions made for the rest of work.

The assumptions made to simplify the model are not justified properly and casts a doubt on the consistency and accuracy of the results. Unfortunately, there is no logical and rationale base to believe in this work at this stage and a major revision is needed. A complete review of the most recent data which have changed our understanding of the OoC mechanics is needed prior to developing a new model based on assumptions and hypotheses that are not valid anymore.

Some minor comments:

The line numbering is not good to make a clear address to a specific section of the text. below, the referencing was made by the page (P) number, line (L) number, left or right (L/R) column (C).

P1, L11, LC: 'built'

P1, L13, LC: <the> motion

P1, L14, LC: 'anatomical'

P1, L17, LC: [, can nevertheless]

P1, L24, LC: What is 'critical oscillator behaviour'?

P1, L29, LC: The model does not 'provide a mechanism', it explains, predicts or an existing mechanism.

First sentence of Introduction: This does not convey any info just by stating the obvious and referring to bunch of papers

Reference 4 is broken

Page 1, Line 36: please rewrite the sentence. It does not follow logical the cause and effect chain. For example, the middle ear does not use the acoustic input to shake the fluid, it passively transfers the vibrations.

P1, L39, LC: 'The wavelength of sound in perilymph is longer than the entire'. The wavelength is frequency dependent, at frequencies above 10 kHz which are still audible by human, the wavelength is shorter than 34mm, an average cochlear length in human.

P1, L40, LC: What is the 'partitioned structure of the cochlea'? It can refer to the 3 fluid channels or discrete cellular arrangement longitudinally, for example. The rest of the sentence is vague too. What is 'a short segment of the partition'?

P1, L42, LC: 'most of the elastic energy...' is there any evidence for this?

P1, L45, LC: <radial cross-sectional slice>

P1, L49, LC: ...<will be> the output.

P1, L49, LC: it is not clear why in this modelling approach OoC 'does not include the BM'. It is not based on any reason and 'accordingly' does not fit here.

P1, L52, LC: what specific characteristics of basal region is taken into account in this model? geometry, material properties etc.?

P1, L54, LC: 'In the basal region ... where the OoC has the greatest impact on amplification and frequency selectivity'. Do you have any reference for this statement? Why would OoC has less impact on the amplification at the apex?

P1, L57, LC: ... (TM) whereas the IHB is not.

Fig.1 what is the reference for the OoC arrangement? The OHCs have angle with the DC (tilted toward the base) any probably not visible with this angle of the view.

Although it is not to scale, but OHB scale is not proportional at all. Use different color for CP.

The figure does not show rest state in which the OHB are supposed to be vertical but it is not stated in the caption. HC's are a number of cells with different orientation. This figure shows a green curvature as a plate.

P1, L26, RC: use anatomical orientations instead of 'clockwise'.

P1, L26, RC: 'motion of the reticular lamina (RL) has no direct effect on the inclination of the IHB.' Why this is assumed?

P1, L29, RC: 'performance' is a vague term here.

P1, L32, RC: 'We would like to answer questions such as: Why the IHB is not attached to the TM?...' this kind of studies do not answer such questions. The aim of this study should be explaining and identifying the benefits of such structures.

P1, L39, RC: 'Many theoretical treatments fall into an extreme category.' Is very vague and non-informative. It is not clear that authors are comparing their method with previous work.

P1, L40, RC: 'At one extreme ...' what extreme? Which one?

'OoC is [presented] by...'

P1, L45, RC: 'Neither of these approaches enables us to answer'. First of all, the authors defined a wrong question, secondly, why the other methods, specially FEM cannot explain the OoC behaviour?

'with idealised geometry and with as few elements and forces as possible, hoping to capture the features'. 'Hope' is not a scientific term! Any study whether experimental or computational, tries to reduce the previous assumptions and improve a realistic representation to explain a phenomenon.

Page 2, 1st paragraph: frequency tuning and amplitude compressions should be defined first.

P2, L9, LC: this kind of referencing is neither scientific nor informative.

P2, L12, LC: what is the purpose of this sentence and bunch of references?

P2, L22, LC: 'Another salient difference is that the RL is not regarded as a completely rigid body' Is the RL rigid in your model or the others? why should it be rigid?

P2, L37, LC: 'but we believe that the important fact is that...'. what makes you believe this?

P2, L40, LC: Most recent measurements contradict this. E.g., 'Ren, T., He, W., & Kemp, D. (2016). Reticular lamina and basilar membrane vibrations in living mouse cochleae. Proceedings of the National Academy of Sciences, 113(35), 9910-9915.'

P2, L50, LC: [showed that the x-...]

P2, L51, LC: This is not correct. Ref. 20 (Fig.2) shows that x&y motions depend on the measurement location and the frequency.

'which is usually disregarded' references for both assumptions are required.

'Pressure exerted by endolymph on OoC components is expected' why? Your rationale or a reference?

Review form: Reviewer 2

Is the manuscript scientifically sound in its present form?

Yes

Are the interpretations and conclusions justified by the results?

Yes

Is the language acceptable?

Yes

Do you have any ethical concerns with this paper?

No

Have you any concerns about statistical analyses in this paper?

No

Recommendation?

Major revision is needed (please make suggestions in comments)

Comments to the Author(s)

While the author's ability to construct the model and get it to be stable is commendable, contribution to new methodology is not strong. Would like to see more detailed comparison with existing data, which is what most cochlear modelers would be interested, and this is actually the advantage of a numerical model. Fig1. The schematic drawing does not reflect the anatomical features in the OC. The Hensen cells seem to have a free end which is not true. The motion the BM is due to pressure difference between the scala media and scala tympani (ST), but I do not see ST in Fig 1. "There are normally three rows of outer hair cells, but we believe that the important fact is that there is more than one, and include just two outer hair cells in our explicit models." But the BM displacement shape is complicated and two OHCs may not address this feature well.

Decision letter (RSOS-200675.R0)

Dear Dr Berger

The Editors assigned to your paper RSOS-200675 "A flexible anatomic set of mechanical models for the organ of Corti" have made a decision based on their reading of the paper and any comments received from reviewers.

Regrettably, in view of the reports received, the manuscript has been rejected in its current form. However, a new manuscript may be submitted which takes into consideration these comments.

We invite you to respond to the comments supplied below and prepare a resubmission of your manuscript. Below the referees' and Editors' comments (where applicable) we provide additional requirements. We provide guidance below to help you prepare your revision.

Please note that resubmitting your manuscript does not guarantee eventual acceptance, and we do not generally allow multiple rounds of revision and resubmission, so we urge you to make every effort to fully address all of the comments at this stage. If deemed necessary by the Editors, your manuscript will be sent back to one or more of the original reviewers for assessment. If the original reviewers are not available, we may invite new reviewers.

Please resubmit your revised manuscript and required files (see below) no later than 14-Feb-2021. Note: the ScholarOne system will 'lock' if resubmission is attempted on or after this deadline. If you do not think you will be able to meet this deadline, please contact the editorial office immediately.

Please note article processing charges apply to papers accepted for publication in Royal Society Open Science (<https://royalsocietypublishing.org/rsos/charges>). Charges will also apply to papers transferred to the journal from other Royal Society Publishing journals, as well as papers submitted as part of our collaboration with the Royal Society of Chemistry (<https://royalsocietypublishing.org/rsos/chemistry>). Fee waivers are available but must be requested when you submit your manuscript (<https://royalsocietypublishing.org/rsos/waivers>).

Thank you for submitting your manuscript to Royal Society Open Science and we look forward to receiving your resubmission. If you have any questions at all, please do not hesitate to get in touch.

on behalf of Pietro Cicuta (Subject Editor)
openscience@royalsociety.org

Reviewer comments to Author:

Reviewer: 1

Comments to the Author(s)

This study aims to develop a simplified analytical model to study the OoC behaviour. The text needs to be reviewed by a person fluent in English. In addition, the text needs to be revised structurally. For example, some terms used in the text is not scientific (e.g., hoping to find... instead of making a hypothesis), referencing is not adequate at all, the abstract looks like part of the introduction and does not convey the summary of the work adequately. The introduction section is a blend of abstract, methods and conclusions. The introduction section does not state the current knowledge and most up-to-date findings that should be used as a base for most of the assumptions made for the rest of work.

The assumptions made to simplify the model are not justified properly and casts a doubt on the consistency and accuracy of the results. Unfortunately, there is no logical and rationale base to believe in this work at this stage and a major revision is needed. A complete review of the most recent data which have changed our understanding of the OoC mechanics is needed prior to developing a new model based on assumptions and hypotheses that are not valid anymore.

Some minor comments:

The line numbering is not good to make a clear address to a specific section of the text. below, the referencing was made by the page (P) number, line (L) number, left or right (L/R) column (C).

P1, L11, LC: 'built'

P1, L13, LC: motion

P1, L14, LC: 'anatomical'

P1, L17, LC: [, can nevertheless]

P1, L24, LC: What is 'critical oscillator behaviour'?

P1, L29, LC: The model does not 'provide a mechanism', it explains, predicts or an existing mechanism.

First sentence of Introduction: This does not convey any info just by stating the obvious and referring to bunch of papers

Reference 4 is broken

Page 1, Line 36: please rewrite the sentence. It does not follow logical the cause and effect chain. For example, the middle ear does not use the acoustic input to shake the fluid, it passively transfers the vibrations.

P1, L39, LC: 'The wavelength of sound in perilymph is longer than the entire'. The wavelength is frequency dependent, at frequencies above 10 kHz which are still audible by human, the wavelength is shorter than 34mm, an average cochlear length in human.

P1, L40, LC: What is the 'partitioned structure of the cochlea'? It can refer to the 3 fluid channels or discrete cellular arrangement longitudinally, for example. The rest of the sentence is vague too. What is 'a short segment of the partition'?

P1, L42, LC: 'most of the elastic energy...' is there any evidence for this?

P1, L45, LC:

P1, L49, LC: ... the output.

P1, L49, LC: it is not clear why in this modelling approach OoC 'does not include the BM'. It is not based on any reason and 'accordingly' does not fit here.

P1, L52, LC: what specific characteristics of basal region is taken into account in this model? geometry, material properties etc.?

P1, L54, LC: 'In the basal region ... where the OoC has the greatest impact on amplification and frequency selectivity'. Do you have any reference for this statement? Why would OoC has less impact on the amplification at the apex?

P1, L57, LC: ... (TM) whereas the IHB is not.

Fig.1 what is the reference for the OoC arrangement? The OHCs have angle with the DC (tilted toward the base) any probably not visible with this angle of the view.

Although it is not to scale, but OHB scale is not proportional at all. Use different color for CP.

The figure does not show rest state in which the OHB are supposed to be vertical but it is not stated in the caption. HC's are a number of cells with different orientation. This figure shows a green curvature as a plate.

P1, L26, RC: use anatomical orientations instead of 'clockwise'.

P1, L26, RC: 'motion of the reticular lamina (RL) has no direct effect on the inclination of the IHB.' Why this is assumed?

P1, L29, RC: 'performance' is a vague term here.

P1, L32, RC: 'We would like to answer questions such as: Why the IHB is not attached to the TM?...' this kind of studies do not answer such questions. The aim of this study should be explaining and identifying the benefits of such structures.

P1, L39, RC: 'Many theoretical treatments fall into an extreme category.' Is very vague and non-informative. It is not clear that authors are comparing their method with previous work.

P1, L40, RC: 'At one extreme ...' what extreme? Which one?

'OoC is [presented] by...'

P1, L45, RC: 'Neither of these approaches enables us to answer'. First of all, the authors defined a wrong question, secondly, why the other methods, specially FEM cannot explain the OoC behaviour?

'with idealised geometry and with as few elements and forces as possible, hoping to capture the features'. 'Hope' is not a scientific term! Any study whether experimental or computational, tries to reduce the previous assumptions and improve a realistic representation to explain a phenomenon.

Page 2, 1st paragraph: frequency tuning and amplitude compressions should be defined first.

P2, L9, LC: this kind of referencing is neither scientific nor informative.

P2, L12, LC: what is the purpose of this sentence and bunch of references?

P2, L22, LC: 'Another salient difference is that the RL is not regarded as a completely rigid body' Is the RL rigid in your model or the others? why should it be rigid?

P2, L37, LC: 'but we believe that the important fact is that...' what makes you believe this?

P2, L40, LC: Most recent measurements contradict this. E.g., 'Ren, T., He, W., & Kemp, D. (2016). Reticular lamina and basilar membrane vibrations in living mouse cochleae. Proceedings of the National Academy of Sciences, 113(35), 9910-9915.'

P2, L50, LC: [showed that the x-...]

P2, L51, LC: This is not correct. Ref. 20 (Fig.2) shows that x&y motions depend on the measurement location and the frequency.

'which is usually disregarded' references for both assumptions are required.

'Pressure exerted by endolymph on OoC components is expected' why? Your rationale or a reference?

Reviewer: 2

Comments to the Author(s)

While the author's ability to construct the model and get it to be stable is commendable, contribution to new methodology is not strong. Would like to see more detailed comparison with existing data, which is what most cochlear modelers would be interested, and this is actually the advantage of a numerical model. Fig1. The schematic drawing does not reflect the anatomical features in the OC. The Hensen cells seem to have a free end which is not true. The motion the BM is due to pressure difference between the scala media and scala tympani (ST), but I do not see ST in Fig 1. "There are normally three rows of outer hair cells, but we believe that the important fact is that there is more than one, and include just two outer hair cells in our explicit models." But the BM displacement shape is complicated and two OHCs may not address this feature well.

===PREPARING YOUR MANUSCRIPT===

===PREPARING YOUR REVISION IN SCHOLARONE===

Author's Response to Decision Letter for (RSOS-200675.R0)

See Appendix A.

RSOS-210016.R0

Review form: Reviewer 1

Is the manuscript scientifically sound in its present form?

No

Are the interpretations and conclusions justified by the results?

Yes

Is the language acceptable?

Yes

Do you have any ethical concerns with this paper?

No

Have you any concerns about statistical analyses in this paper?

No

Recommendation?

Accept with minor revision (please list in comments)

Comments to the Author(s)

General comments and suggestions:

The presentation of the work has been improved significantly. Authors make a clever use of dimensionless analysis, although at the end I would suggest scaling the frequency and spatial dimensions to a specific specimen (animal or human) to provide a realistic perception of the numbers (e.g., critical frequency). The unsynchronized work generated by the OHCs are assumed to be the reason for lower vibrations, but I could not distinguish it from mechanical energy dissipation caused by the mechanical and passive damping of the system. The amplification mechanism for the slice model is thoroughly studied, but extrapolating it to the full-length cochlea needs a continuous segmentation or at least wave-dependent repetition of the slice model. For example, the amplification of the OHCs shifts the resonance to higher frequencies e.g., in fig.2M of (Lee, H. Y., Raphael, P. D., Xia, A., Kim, J., Grillet, N., Applegate, B. E., ... & Oghalai, J. S. (2016). Two-dimensional cochlear micromechanics measured in vivo demonstrate radial tuning within the mouse organ of Corti. *Journal of Neuroscience*, 36(31), 8160-8173.) the resonance peak has shifted from 5kHz at 80 dB to 10 kHz at 10 dB. It was not clear to me that model can predict the shift to make conclusion about the amplification of the traveling wave.

Specific comments:

Maybe it is the RSOS style, but the line numbering is still troubling.

P1,L38, RC: 'typical audible frequencies' is a spectrum from 20 to 20,000Hz, the wavelength varies by 3 order of magnitudes.

P1,L40, RC: First you refer to <fluid> chambers as 'partitions', then in the same sentence partition is referring to a small cross-sectional region of the BM where the "energy" is

“deposited”. The switch between reference of the partition continues in the next sentences. The definition of ‘partition’ could be different as mentioned in different papers (fluid chambers partitions, cross-sectional partitions of the cochlea etc.), but should be consistent in the text.

L55: ‘separate collagen fibers, with length, width and stiffness that gradually vary’ could you put a reference for this? I believe the collagen fibers vary in concentration but not in width and length through the cochlear length (Cabezudo, L. M. (1978). The ultrastructure of the basilar membrane in the cat. *Acta oto-laryngologica*, 86(1-6), 160-175.)

LC, L11: Please rewrite this sentence: ‘exposed only to pressure differences within the SM, is exposed to the large pressure difference between’.

L22: ‘frequency tuning (output sharply peaked at some frequency for a given input) and amplitude compression (input changes by several orders of magnitude give rise to significantly smaller changes of the output).’ What are inputs and outputs? I assume the input is the mechanical excitation that is delivered to the cochlea at the oval window and the output is electrical signals of the auditory neurons. Please revise the definition of the compression.

L28: what does ‘theoretical treatments’ mean?

L29: the representation of mechanical systems with electrical circuit models is well-accepted method in mechanics of hearing. The Newton law is a simple relationship between force and acceleration which analogously could be represented by Maxwell’s equations in electrical systems. I am a mechanical engineer, but I don’t think this is an issue to question the method.

L33: this is not true, there are FE models in which the cochlear amplification and nonlinearity have been implemented (e.g., Motallebzadeh, H., Soons, J. A., & Puria, S. (2018). Cochlear amplification and tuning depend on the cellular arrangement within the organ of Corti. *Proceedings of the National Academy of Sciences*, 115(22), 5762-5767.). Any modelling study is basically a trade of simplicity and realistic representation of a system. Even with detailed FE models, one can differentiate contribution of different components on the overall response. When you make a realistic system the interactions between the components get more complicated, but the model is still following the physical governing laws, you make a simple model, you neglect more details, and deviate from the realistic behaviour and you may not see the features that simplification cause, but what you see is easier to understand. This paragraph is not a strong argument to question other methods to advertise your method.

L43: ‘Substantial evidence(s) has(have) led ...

L57: In recent models (full length cochlear models) the pressure is a function of place and time as well (e.g., Sasmal, A., & Grosh, K. (2019). Unified cochlear model for low-and high-frequency mammalian hearing. *Proceedings of the National Academy of Sciences*, 116(28), 13983-13988. Motallebzadeh, H., Soons, J. A., & Puria, S. (2018). Cochlear amplification and tuning depend on the cellular arrangement within the organ of Corti. *Proceedings of the National Academy of Sciences*, 115(22), 5762-5767.). Do you refer any specific work that considers the RL as a rigid body? I can think of few paper (e.g., Lim, K. M., & Steele, C. R. (2002). A three-dimensional nonlinear active cochlear model analyzed by the WKB-numeric method. *Hearing research*, 170(1-2), 190-205.) in which this assumption is made to find low-frequency behaviour or force transmissions within the OoC.

P3, L14, RC: Do you have any specific reference for this fact that the R number is very small? ‘rotational equation of motion’  ‘momentum equation’ (motion equations refer to kinematics and momentum equations refer to the dynamics or force balance)
‘equation of motion’  ‘momentum equation’

P3, L33: what is Greek letter ‘v’ in this term? Is it velocity, or volume velocity or something else? Is it the same letter in Equation 1 (there are two Greek letters similar to v)? Is yes, please be consistent.

P4, L37, RC: There are more recent measurements on the TM mechanical properties (e.g., Sellon, J. B., Ghaffari, R., & Freeman, D. M. (2018). The tectorial membrane: mechanical properties and functions. *Cold Spring Harbor perspectives in medicine*, a033514.)

Equation 7: Please define ‘sgn’ function

P4, L52: please elaborate 'work performed by the bundle motility vanishes'. Work is performed by force do you mean that hair bundles, in addition to the OHC body, generate force?

Line 27: Please elaborate on 'non vanishing work'. Damped or dissipated energy refers to energy loss due to the damping or friction but here I believe you are referring to the work due to the OHC motility.

P5, L11: remind the reader that the length 1 is in the adopted normalized coordinate system.

L46: please elaborate on 'we do not consider noise that arises in the OoC itself'. What kind of noise the OoC can produce and what exactly noise mean here? Do you mean perturbations due to the nonuniform distribution of the mechanical properties?

'A. Main Results' is not a specific subtitle to differentiate from the rest of the results. Maybe 'summary of key findings'?

L 37, LC: please elaborate on 'self-oscillations (non zero output for zero input)'. Do you mean the system is unstable or underdamped in this specific satiation?

P5, L42, RC: Could you translate the units and dimensions for a specific specimen so the reader can have an idea of the critical frequency? 'corresponds to a contraction of a few percent' of a dimensionless parameter is not enough to evaluate its value.

L52, RC: 'then the OoC would provide additional tuning; if it is not, the OoC would provide an alternative mechanism for tuning' what is the difference of 'additional tuning' and 'alternative mechanism for tuning'?

P8, L53 LC: 'is in the range between 0° and 180° (or equivalent)' please clarify. What is equivalent to $0-180^\circ$? It could be either π plus/minus or 2π plus/minus, depending on the periodicity function.

P8, L41 RC: 'Within a more realistic model, the energy exchange described here should be regarded as a contribution.' Contribution to?

P8, L43 RC: The attenuation according to the calculations here, is coming from unsynchronized OHC force, but is there any direct calculation of the attenuations due to the damping of the system? Is it possible even when OHC provides constructive force (and work) that work is dissipated due to the mechanical damping?

P10, 5, LC: although HC's are outside the subreticular space, but do to their large deformations (because of lower stiffness), could they also pump fluid into the subreticular space?

Review form: Reviewer 2

Is the manuscript scientifically sound in its present form?

No

Are the interpretations and conclusions justified by the results?

No

Is the language acceptable?

No

Do you have any ethical concerns with this paper?

No

Have you any concerns about statistical analyses in this paper?

No

Recommendation?

Major revision is needed (please make suggestions in comments)

Comments to the Author(s)

Some of the assumptions made to the model are not justified properly and lack necessary supporting materials, which makes it difficult to follow the logic of the work. In the abstract and throughout the text, verb tense was used incorrectly.

Some other comments can be found below:

To explain comments clearly, the referencing was made by the page (P) number, line (L) number, and left or right (L/R) column.

P1, L38~40, L: "For typical audible frequencies, the wave length of sound in the perilymph is of the same order of magnitude as the length of the entire cochlea." This refers to the human cochlea, but there is no clear explanation that the model proposed in this paper was for the human or other species.

P1, L55, L: the BM thickness also varies along its length.

P1, L52, L: "The partition is composed of the BM, the organ of Corti (OoC) and the tectorial membrane (TM)". This description includes only three main components, but there are other important components within the cochlear partition.

P1, L57, L: "The BM has a large Young modulus", the term Young modulus is usually used in the cochlear model but not for describing the physiological feature of the BM.

P1, L11, R: The statement "pressure differences within the SM" is not proper, there is a pressure difference between the SM and ST, what does pressure difference within the SM refer to? There is another conflict with this point (P2, L25~L27, R) where the pressure in the SM will be taken equal to the pressure in the tissues under the RL and the HC.

P2, L44, R: "width" should be changed to "thickness".

P2, L3, L: Is there any supporting evidence?

P2, L19~23, L: It would be interesting to see the difference of using three OHCs.

P2, L26, R: "the RL pivots as a rigid beam around the pillar cells head". This was mentioned several times in the paper and was used as a reference for setting the origin. But this somewhat conflicts with the point that the authors introduced cuticular plates into the model.

P2, L40~42, L: "Accordingly, except for rotational and for fluid motion, motion will be restricted to the y-direction." The authors mentioned that the amplitudes of the x and of the y components of the relative motion are of the same order of magnitude, which implies that motions in both directions are important. (P10, L2~3, L) "According to our models, the fluid flow at the IHB region is driven by the vertical motion of the CPs," This is because the current setting only considers motion in the y-direction.

Decision letter (RSOS-210016.R0)

Dear Dr Berger

The Editors assigned to your paper RSOS-210016 "A flexible anatomical set of mechanical models for the organ of Corti" have now received comments from reviewers and would like you to revise the paper in accordance with the reviewer comments and any comments from the Editors. Please note this decision does not guarantee eventual acceptance.

Please submit your revised manuscript and required files (see below) no later than 21 days from today's (ie 17-May-2021) date. Note: the ScholarOne system will 'lock' if submission of the revision is attempted 21 or more days after the deadline. If you do not think you will be able to meet this deadline please contact the editorial office immediately.

on behalf of Prof Pietro Cicuta (Subject Editor)
openscience@royalsociety.org

Associate Editor Comments to Author:

Please be aware that the revision decision is likely to be your final opportunity to satisfy the reviewers and editors that your paper is ready for acceptance - please ensure you carefully engage with the reviewer queries/comments, and supply not only a fully revised version of your manuscript with the revision (making sure to mark up the changes clearly) but a point-by-point response for the reviewers.

Reviewer comments to Author:

Reviewer: 2

Comments to the Author(s)

Some of the assumptions made to the model are not justified properly and lack necessary supporting materials, which makes it difficult to follow the logic of the work. In the abstract and throughout the text, verb tense was used incorrectly.

Some other comments can be found below:

To explain comments clearly, the referencing was made by the page (P) number, line (L) number, and left or right (L/R) column.

P1, L38~40, L: "For typical audible frequencies, the wave length of sound in the perilymph is of the same order of magnitude as the length of the entire cochlea." This refers to the human cochlea, but there is no clear explanation that the model proposed in this paper was for the human or other species.

P1, L55, L: the BM thickness also varies along its length.

P1, L52, L: "The partition is composed of the BM, the organ of Corti (OoC) and the tectorial membrane (TM)". This description includes only three main components, but there are other important components within the cochlear partition.

P1, L57, L: "The BM has a large Young modulus", the term Young modulus is usually used in the cochlear model but not for describing the physiological feature of the BM.

P1, L11, R: The statement "pressure differences within the SM" is not proper, there is a pressure difference between the SM and ST, what does pressure difference within the SM refer to? There is another conflict with this point (P2, L25~L27, R) where the pressure in the SM will be taken equal to the pressure in the tissues under the RL and the HC.

P2, L44, R: "width" should be changed to "thickness".

P2, L3, L: Is there any supporting evidence?

P2, L19~23, L: It would be interesting to see the difference of using three OHCs.

P2, L26, R: "the RL pivots as a rigid beam around the pillar cells head". This was mentioned several times in the paper and was used as a reference for setting the origin. But this somewhat conflicts with the point that the authors introduced cuticular plates into the model.

P2, L40~42, L: "Accordingly, except for rotational and for fluid motion, motion will be restricted to the y-direction." The authors mentioned that the amplitudes of the x and of the y components of the relative motion are of the same order of magnitude, which implies that motions in both directions are important. (P10, L2~3, L) "According to our models, the fluid flow at the IHB region is driven by the vertical motion of the CPs," This is because the current setting only considers motion in the y-direction.

Reviewer: 1

Comments to the Author(s)

General comments and suggestions:

The presentation of the work has been improved significantly. Authors make a clever use of dimensionless analysis, although at the end I would suggest scaling the frequency and spatial dimensions to a specific specimen (animal or human) to provide a realistic perception of the numbers (e.g., critical frequency). The unsynchronized work generated by the OHCs are assumed to be the reason for lower vibrations, but I could not distinguish it from mechanical energy dissipation caused by the mechanical and passive damping of the system. The amplification mechanism for the slice model is thoroughly studied, but extrapolating it to the full-length cochlea needs a continuous segmentation or at least wave-dependent repetition of the slice model. For example, the amplification of the OHCs shifts the resonance to higher frequencies e.g., in fig.2M of (Lee, H. Y., Raphael, P. D., Xia, A., Kim, J., Grillet, N., Applegate, B. E., ... & Oghalai, J. S. (2016). Two-dimensional cochlear micromechanics measured in vivo demonstrate radial tuning within the mouse organ of Corti. *Journal of Neuroscience*, 36(31), 8160-8173.) the

resonance peak has shifted from 5kHz at 80 dB to 10 kHz at 10 dB. It was not clear to me that model can predict the shift to make conclusion about the amplification of the traveling wave. Specific comments:

Maybe it is the RSOS style, but the line numbering is still troubling.

P1,L38, RC: 'typical audible frequencies' is a spectrum from 20 to 20,000Hz, the wavelength varies by 3 order of magnitudes.

P1,L40, RC: First you refer to chambers as 'partitions', then in the same sentence partition is referring to a small cross-sectional region of the BM where the "energy" is "deposited". The switch between reference of the partition continues in the next sentences. The definition of 'partition' could be different as mentioned in different papers (fluid chambers partitions, cross-sectional partitions of the cochlea etc.), but should be consistent in the text.

L55: 'separate collagen fibers, with length, width and stiffness that gradually vary' could you put a reference for this? I believe the collagen fibers vary in concentration but not in width and length thorough the cochlear length (Cabezudo, L. M. (1978). The ultrastructure of the basilar membrane in the cat. *Acta oto-laryngologica*, 86(1-6), 160-175.)

LC, L11: Please rewrite this sentence: 'exposed only to pressure differences within the SM, is exposed to the large pressure difference between'.

L22: 'frequency tuning (output sharply peaked at some frequency for a given input) and amplitude compression (input changes by several orders of magnitude give rise to significantly smaller changes of the output).' What are inputs and outputs? I assume the input is the mechanical excitation that is delivered to the cochlea at the oval window and the output is electrical signals of the auditory neurons. Please revise the definition of the compression.

L28: what does 'theoretical treatments' mean?

L29: the representation of mechanical systems with electrical circuit models is well-accepted method in mechanics of hearing. The newton law is a simple relationship between force and acceleration which analogously could be represented by Maxwell's equations in electrical systems. I am a mechanical engineer, but I don't think this is an issue to question the method.

L33: this is not true, there are FE models in which the cochlear amplification and nonlinearity have been implemented (e.g., Motallebzadeh, H., Soons, J. A., & Puria, S. (2018). Cochlear amplification and tuning depend on the cellular arrangement within the organ of Corti.

Proceedings of the National Academy of Sciences, 115(22), 5762-5767.). Any modelling study is basically a trade of simplicity and realistic representation of a system. Even with detailed FE models, one can differentiate contribution of different components on the overall response. When you make a realistic system the interactions between the components get more complicated, but the model is still following the physical governing laws, you make a simple model, you neglect more details, and deviate from the realistic behaviour and you may not see the features that simplification cause, but what you see is easier to understand. This paragraph is not a strong argument to question other methods to advertise your method.

L43: 'Substantial evidence(s) has(have) led ...

L57: In recent models (full length cochlear models) the pressure is a function of place and time as well (e.g., Sasmal, A., & Grosh, K. (2019). Unified cochlear model for low- and high-frequency mammalian hearing. *Proceedings of the National Academy of Sciences*, 116(28), 13983-13988.

Motallebzadeh, H., Soons, J. A., & Puria, S. (2018). Cochlear amplification and tuning depend on the cellular arrangement within the organ of Corti. *Proceedings of the National Academy of Sciences*, 115(22), 5762-5767.). Do you refer any specific work that considers the RL as a rigid body? I can think of few paper (e.g., Lim, K. M., & Steele, C. R. (2002). A three-dimensional nonlinear active cochlear model analyzed by the WKB-numeric method. *Hearing research*, 170(1-2), 190-205.) in which this assumption is made to find low-frequency behaviour or force transmissions within the OoC.

P3, L14, RC: Do you have any specific reference for this fact that the R number is very small?

'rotational equation of motion'  'momentum equation' (motion equations refer to kinematics and momentum equations refer to the dynamics or force balance)

'equation of motion'  'momentum equation'

P3, L33: what is Greek letter 'v' in this term? Is it velocity, or volume velocity or something else? Is it the same letter in Equation 1 (there are two Greek letters similar to v)? Is yes, please be consistent.

P4, L37, RC: There are more recent measurements on the TM mechanical properties (e.g., Sellon, J. B., Ghaffari, R., & Freeman, D. M. (2018). The tectorial membrane: mechanical properties and functions. Cold Spring Harbor perspectives in medicine, a033514.)

Equation 7: Please define 'sgn' function

P4, L52: please elaborate 'work performed by the bundle motility vanishes'. Work is performed by force do you mean that hair bundles, in addition to the OHC body, generate force?

Line 27: Please elaborate on 'non vanishing work'. Damped or dissipated energy refers to energy loss due to the damping or friction but here I believe you are referring to the work due to the OHC motility.

P5, L11: remind the reader that the length 1 is in the adopted normalized coordinate system.

L46: please elaborate on 'we do not consider noise that arises in the OoC itself'. What kind of noise the OoC can produce and what exactly noise mean here? Do you mean perturbations due to the nonuniform distribution of the mechanical properties?

'A. Main Results' is not a specific subtitle to differentiate from the rest of the results. Maybe 'summary of key findings'?

L 37, LC: please elaborate on 'self-oscillations (non zero output for zero input)'. Do you mean the system is unstable or underdamped in this specific situation?

P5, L42, RC: Could you translate the units and dimensions for a specific specimen so the reader can have an idea of the critical frequency? 'corresponds to a contraction of a few percent' of a dimensionless parameter is not enough to evaluate its value.

L52, RC: 'then the OoC would provide additional tuning; if it is not, the OoC would provide an alternative mechanism for tuning' what is the difference of 'additional tuning' and 'alternative mechanism for tuning'?

P8, L53 LC: 'is in the range between 0° and 180° (or equivalent)' please clarify. What is equivalent to 0-180? It could be either pi plus/minus or 2pi plus/minus, depending on the periodicity function.

P8, L41 RC: 'Within a more realistic model, the energy exchange described here should be regarded as a contribution.' Contribution to?

P8, L43 RC: The attenuation according to the calculations here, is coming from unsynchronized OHC force, but is there any direct calculation of the attenuations due to the damping of the system? Is it possible even when OHC provides constructive force (and work) that work is dissipated due to the mechanical damping?

P10, 5, LC: although HC's are outside the subtectorial space, but do to their large deformations (because of lower stiffness), could they also pump fluid into the subtectorial space?

===PREPARING YOUR MANUSCRIPT===

Please ensure that you include an acknowledgements' section before your reference list/bibliography. This should acknowledge anyone who assisted with your work, but does not

qualify as an author per the guidelines at <https://royalsociety.org/journals/ethics-policies/openness/>.

===PREPARING YOUR REVISION IN SCHOLARONE===

- Ensure that your data access statement meets the requirements at <https://royalsociety.org/journals/authors/author-guidelines/#data>. You should ensure that you cite the dataset in your reference list. If you have deposited data etc in the Dryad repository, please include both the 'For publication' link and 'For review' link at this stage.
- If you are requesting an article processing charge waiver, you must select the relevant waiver option (if requesting a discretionary waiver, the form should have been uploaded at Step 3 'File upload' above).
- If you have uploaded ESM files, please ensure you follow the guidance at <https://royalsociety.org/journals/authors/author-guidelines/#supplementary-material> to include a suitable title and informative caption. An example of appropriate titling and captioning may be found at https://figshare.com/articles/Table_S2_from_Is_there_a_trade-off_between_peak_performance_and_performance_breadth_across_temperatures_for_aerobic_scope_in_teleost_fishes_/3843624.

Author's Response to Decision Letter for (RSOS-210016.R0)

See Appendix B.

RSOS-210016.R1 (Revision)

Review form: Reviewer 1

Is the manuscript scientifically sound in its present form?

Yes

Are the interpretations and conclusions justified by the results?

Yes

Is the language acceptable?

Yes

Do you have any ethical concerns with this paper?

No

Have you any concerns about statistical analyses in this paper?

No

Recommendation?

Accept with minor revision (please list in comments)

Comments to the Author(s)

Minor comments:

P1,4, RC: "The BM acts as an elastic barrier and is exposed to the large pressure difference between the SM and the ST", the pressure gradient between SM and ST varies along the length as a function of frequency. Apical to the characteristic frequency there is almost no pressure difference between these two chambers.

P2 26-46, LC: In my opinion the main feature of this work in implementing analytical approach to solve the mechanics of the OoC. FE studies use numerical discretion methods and lumped element models use electrical analogy.

P2 2-4 RC: RL and CPs are making a continues frame, which is flexible at CPs and rigid at RL segments. Authors emphasize on the RL characteristics 'A salient' feature of their method and a significant deviation from 'featureless body' from the whole studies in the literature. There are studies of both rigid and flexible RL and I suggest to emphasize on more important characteristics on the study.

P3 14-15 RC: "fluid velocity as being similar to that of the BM velocity". Due to the no-slip boundary condition of the NS equations on the BM and fluid interface, their velocities are exactly the same (not similar) at the BM surface.

P5 51, RC: Is this instability part of the OoC physiology which results in spontaneous otoacoustic emissions?

P6 19 LC: The hearing frequency is more common than angular frequency (ω) in hearing acoustics, although they are linearly related, but I would suggest converting them to actual frequency (as done in line 12 the same page).

P9 F. Nonlinearity: I could not follow the argument of nonlinearity due to the non-harmonic theta. What kind of nonlinearity is referred here? How does non-sinusoidal theta and the Fourier transfer of it establish nonlinearity?

Decision letter (RSOS-210016.R1)

Dear Dr Berger

On behalf of the Editors, we are pleased to inform you that your Manuscript RSOS-210016.R1 "A flexible anatomical set of mechanical models for the organ of Corti" has been accepted for publication in Royal Society Open Science subject to minor revision in accordance with the referees' reports. Please find the referees' comments along with any feedback from the Editors below my signature.

Please submit your revised manuscript and required files (see below) no later than 7 days from today's (ie 09-Aug-2021) date. Note: the ScholarOne system will 'lock' if submission of the revision is attempted 7 or more days after the deadline. If you do not think you will be able to meet this deadline please contact the editorial office immediately.

Please note article processing charges apply to papers accepted for publication in Royal Society Open Science (<https://royalsocietypublishing.org/rsos/charges>). Charges will also apply to papers transferred to the journal from other Royal Society Publishing journals, as well as papers

submitted as part of our collaboration with the Royal Society of Chemistry (<https://royalsocietypublishing.org/rsos/chemistry>). Fee waivers are available but must be requested when you submit your revision (<https://royalsocietypublishing.org/rsos/waivers>).

on behalf of Pietro Cicuta (Subject Editor)
openscience@royalsociety.org

Reviewer comments to Author:
Reviewer: 1

Comments to the Author(s)

Minor comments:

P1,4, RC: "The BM acts as an elastic barrier and is exposed to the large pressure difference between the SM and the ST", the pressure gradient between SM and ST varies along the length as a function of frequency. Apical to the characteristic frequency there is almost no pressure difference between these two chambers.

P2 26-46, LC: In my opinion the main feature of this work in implementing analytical approach to solve the mechanics of the OoC. FE studies use numerical discretion methods and lumped element models use electrical analogy.

P2 2-4 RC: RL and CPs are making a continues frame, which is flexible at CPs and rigid at RL segments. Authors emphasize on the RL characteristics 'A salient' feature of their method and a significant deviation from 'featureless body' from the whole studies in the literature. There are studies of both rigid and flexible RL and I suggest to emphasize on more important characteristics on the study.

P3 14-15 RC: "fluid velocity as being similar to that of the BM velocity". Due to the no-slip boundary condition of the NS equations on the BM and fluid interface, their velocities are exactly the same (not similar) at the BM surface.

P5 51, RC: Is this instability part of the OoC physiology which results in spontaneous otoacoustic emissions?

P6 19 LC: The hearing frequency is more common that angular frequency (ω) in hearing acoustics, although they are linearly related, but I would suggest converting them to actual frequency (as done in line 12 the same page).

P9 F. Nonlinearity: I could not follow the argument of nonlinearity due to the non-harmonic theta. What kind of nonlinearity is referred here? How does non-sinusoidal theta and the Fourier transfer of it establish nonlinearity?

===PREPARING YOUR MANUSCRIPT===

===PREPARING YOUR REVISION IN SCHOLARONE===

- If you are providing image files for potential cover images, please upload these at this step, and inform the editorial office you have done so. You must hold the copyright to any image provided.
- A copy of your point-by-point response to referees and Editors. This will expedite the preparation of your proof.

- Ensure that your data access statement meets the requirements at <https://royalsociety.org/journals/authors/author-guidelines/#data>. You should ensure that you cite the dataset in your reference list. If you have deposited data etc in the Dryad repository, please only include the 'For publication' link at this stage. You should remove the 'For review' link.
- If you are requesting an article processing charge waiver, you must select the relevant waiver option (if requesting a discretionary waiver, the form should have been uploaded at Step 3 'File upload' above).
- If you have uploaded ESM files, please ensure you follow the guidance at <https://royalsociety.org/journals/authors/author-guidelines/#supplementary-material> to include a suitable title and informative caption. An example of appropriate titling and captioning may be found at [https://figshare.com/articles/Table_S2_from_Is_there_a_trade-off_between_peak_performance_and_performance_breadth_across_temperatures_for_aerobic_sc ope_in_teleost_fishes_/3843624](https://figshare.com/articles/Table_S2_from_Is_there_a_trade-off_between_peak_performance_and_performance_breadth_across_temperatures_for_aerobic_scope_in_teleost_fishes_/3843624).

Author's Response to Decision Letter for (RSOS-210016.R1)

See Appendix C.

Decision letter (RSOS-210016.R2)

Dear Dr Berger,

I am pleased to inform you that your manuscript entitled "A flexible anatomical set of mechanical models for the organ of Corti" is now accepted for publication in Royal Society Open Science.

on behalf of Pietro Cicuta (Subject Editor)
openscience@royalsociety.org

Appendix A

Dear Editor,

Thank you for your mail from 7-Aug-2020 Re: our manuscript, RSOS-200675, entitled "A flexible anatomical set of mechanical models for the organ of Corti".

We carefully read the comments, corrections and suggestions of the reviewers and amended the paper accordingly. We list below in bold face font our reply to each of the reviewer comments and requests. In addition to addressing these specific points, we made additional changes and improvements throughout the paper following the general philosophy expressed by the reviewers.

We thank the reviewers for their careful reading of the paper and for their productive criticism that has helped us improve the current version significantly. We hope that the revised paper can now be accepted for publication.

Concerning the style of our Introduction, the author guidelines of the Royal Society state that the authors should feel free to subdivide the main text as best suits the article. We followed the standard format in our disciplines, i.e, the first section of the article is the Introduction, which customarily contains a brief review of the background (complemented by a cluster of references that provide the additional required background details), states the goals of the study at hand, cites studies with similar goals, and describes the differences between these previous studies and the one at hand.

Due to the unconventional character of our goal, we took care to not just state what it is, but also what it is not. We did not look for a better description of the OoC in some particular animal. Instead, we built an instrument that enables any researcher to try any particular hypothesis on the mechanical behaviour of some anatomical component of the OoC, and assess the influence that it has on the mechanical activity of OoC as a whole. This being the case, there is room in our article for revisiting hypotheses that have not been experimentally supported. We are confident that the instrument that we provide can be useful to the readers of RSOS.

Answers to the reviewers

Note that reference numbers indicated below correspond to those in the present version of the manuscript.

Replies to Reviewer 1

Reviewer: This study aims to develop a simplified analytical model to study the OoC behaviour. The text needs to be reviewed by a person fluent in English. In addition, the text needs to be revised structurally. For example, some terms used in the text is not scientific (e.g., hoping to find... instead of making a hypothesis), referencing is not adequate at all, the abstract looks like part of the introduction and does not convey the summary of the work adequately. The introduction section is a blend of abstract, methods and conclusions. The introduction section does not state the current knowledge and most up-to-date findings that should be used as a base for most of the assumptions made for the rest of work. The assumptions made to simplify the model are not justified properly and casts a doubt on the consistency and accuracy of the results. Unfortunately, there is no logical and rationale base to believe in this work at this stage and a major revision is needed. A complete review of the most recent data

which have changed our understanding of the OoC mechanics is needed prior to developing a new model based on assumptions and hypotheses that are not valid anymore.

Authors: We amended the text in many places according to the reviewer comments, suggestions, and general criticism. Changes made in direct response to specific reviewer's comments are detailed below. In the modified text of the paper we highlighted these specific changes as well as those made following the spirit of the reviewers' feedback.

The abstract has been rewritten to convey a concise summary of the work we present in the paper. The paper was professionally reviewed by a language editor and was corrected accordingly. We replaced phrases that Reviewer 1 found as "non scientific" by more precise phrases throughout. To the best of our knowledge, we have comprehensively cited the literature which we consider relevant to the present study, either directly or through some review. The assumptions made in our work are based either on the literature we quote or on the spirit of this study, which favors taking the simplest possible model (what is usually dubbed "the minimal model").

We also carefully followed all the reviewer's suggestions regarding the justification of the models and methodology which we now set forward in the Introduction section.

Some minor comments:

The line numbering is not good to make a clear address to a specific section of the text. below, the referencing was made by the page (P) number, line (L) number, left or right (L/R) column (C).

P1, L11, LC: 'built'

The text was corrected.

P1, L13, LC: motion

Corrected in the new abstract.

P1, L14, LC: 'anatomical'

The text was corrected.

P1, L17, LC: [, can nevertheless]

The abstract was rewritten, and this sentence was removed.

P1, L24, LC: What is 'critical oscillator behaviour'?

This is explained in the main body of the paper (Section 4A and Appendix 3). We are not accustomed to including citations in the abstract.

P1, L29, LC: The model does not 'provide a mechanism', it explains, predicts or an existing mechanism.

The abstract was rewritten, and this term was removed.

First sentence of Introduction: This does not convey any info just by stating the obvious and referring to bunch of papers

We think such an opening sentence might be useful to a reader not familiar with the basic literature. In fact, this is a common practice. For instance, the first sentence in a recent paper in RSOS (

<https://royalsocietypublishing.org/doi/10.1098/rsos.200527>) reads: "*Molecular motors that facilitate intracellular transport can move cargoes such as vesicles, lipid droplets or mitochondria [1–7].*"

Reference 4 is broken

The reference was corrected in the new text.

Page 1, Line 36: please rewrite the sentence. It does not follow logical the cause and effect chain. For example, the middle ear does not use the acoustic input to shake the fluid, it passively transfers the vibrations.

We replaced "used by the middle ear to shake perilymph in the inner ear, while" by "passed by the middle ear to the perilymph in the inner ear, while"

P1, L39, LC: 'The wavelength of sound in perilymph is longer than the entire'. The wavelength is frequency dependent, at frequencies above 10 kHz which are still audible by human, the wavelength is shorter than 34mm, an average cochlear length in human.

We now write: "For typical audible frequencies, the wavelength of sound in the perilymph is of the same order of magnitude as the length of the entire..."

P1, L40, LC: What is the 'partitioned structure of the cochlea'? It can refer to the 3 fluid channels or discrete cellular arrangement longitudinally, for example. The rest of the sentence is vague too. What is 'a short segment of the partition'?

In the new text we now write: "...partitioned structure of the cochlea [in which the basilar membrane (BM) responds to the pressure difference between the chambers at each of its sides] gives rise to a travelling surface wave with shrinking wavelength (8, 9), such that the energy deposited on the partition is concentrated in a segment shorter than a millimeter (10). The partitioned structure of the cochlea is described, e.g., in (4,6). Within the cochlea a "horizontal" partition divides between (i) the scala media (SM), filled with endolymph, and further "above" it the scala vestibuli, filled with perilymph, and (ii) the scala tympani, which is connected to the scala vestibuli at the apex of the cochlea. The partition is composed of the BM, the organ of Corti (OoC) and the tectorial membrane (TM). The BM is made up of separate collagen fibers, with length, width and stiffness that gradually vary from the base to the apex of the cochlea."

P1, L42, LC: 'most of the elastic energy...' is there any evidence for this?

We added "The BM has a large Young modulus, and, as opposed to the rest of the partition that is exposed only to pressure differences within the SM, is exposed to the large pressure difference between the SM and the scala tympani. As such, most of the elastic..."

P1, L49, LC: ... the output.

The text was corrected.

P1, L49, LC: it is not clear why in this modelling approach OoC 'does not include the BM'. It is not based on any reason and 'accordingly' does not fit here.

This sentence was removed.

P1, L52, LC: what specific characteristics of basal region is taken into account in this model? geometry, material properties etc.?

We added a reference that expands on this point.

P1, L54, LC: 'In the basal region ... where the OoC has the greatest impact on amplification and frequency selectivity'. Do you have any reference for this statement? Why would OoC has less impact on the amplification at the apex?

A reference is now provided.

P1, L57, LC: ... (TM) whereas the IHB is not.

We corrected this sentence in the new text.

Fig.1 what is the reference for the OoC arrangement? The OHCs have angle with the DC (tilted toward the base) any probably not visible with this angle of the view.
Although it is not to scale, but OHB scale is not proportional at all. Use different color for CP.
The figure does not show rest state in which the OHB are supposed to be vertical but it is not stated in the caption. HC's are a number of cells with different orientation. This figure shows a green curvature as a plate.

We have extended the caption in Fig. 1 and added several references. We believe that these additions settle all the misunderstandings.

P1, L26, RC: use anatomical orientations instead of 'clockwise'.

We now write in the text: "...OHB tilts in the excitatory direction (clockwise)"

P1, L26, RC: 'motion of the reticular lamina (RL) has no direct effect on the inclination of the IHB.' Why this is assumed?

This statement has been sharpened. We now write "We neglect the influence of the inclination of the reticular lamina (RL) on the inclination of the IHB"

P1, L29, RC: 'performance' is a vague term here.

This paragraph was rewritten as highlighted in the reply to the next issue immediately below.

P1, L32, RC: 'We would like to answer questions such as: Why the IHB is not attached to the TM?...' this kind of studies do not answer such questions. The aim of this study should be explaining and identifying the benefits of such structures.

We rewrote this paragraph as follows: "Our aspiration is not necessarily to obtain a precise description of the mechanical parameters of the different components of the OoC, but rather to gain insight into how these components cooperate to achieve its global operation. In particular, we would like to provide possible explanations for the benefits of having IHBs that are not attached to the TM, and of the curious fact that after transforming fluid flow into mechanical vibration, this vibration is transformed back into fluid flow, this time along a narrow channel, involving high dissipation. Other questions we would like to pursue include what is the advantage of having several OHCs, rather than a single stronger OHC, how does an OHC perform mechanical work on the system, and whether there is any role to passive components such as the Hensen cells (HC)."

P1, L39, RC: 'Many theoretical treatments fall into an extreme category.' Is very vague and non-informative. It is not clear that authors are comparing their method with previous work.

Indeed, we do not make a direct comparison. The purpose of this paragraph is to point out that our goal is different from that of many other published works, and we therefore require a different methodology.

P1, L40, RC: 'At one extreme ...' what extreme? Which one?

This is now clarified in the amended text.

P1, L45, RC: 'Neither of these approaches enables us to answer'. First of all, the authors defined a wrong question, secondly, why the other methods, specially FEM cannot explain the OoC behaviour?

We now write "The models in (12, 13, 15–17) are linear and thus cannot handle inherently nonlinear effects such as bifurcations. Moreover, when a huge number of degrees of freedom are involved, the task of identifying a central feature that brings about a given behaviour becomes less transparent."

'with idealised geometry and with as few elements and forces as possible, hoping to capture the features'. 'Hope' is not a scientific term! Any study whether experimental or computational, tries to reduce the previous assumptions and improve a realistic representation to explain a phenomenon.

The text was corrected. As stated in the paper, we are not looking for a representation of a measured phenomenon, but rather building an instrument that can qualitatively test the relation between sets of assumptions and OoC responses.

Page 2, 1st paragraph: frequency tuning and amplitude compressions should be defined first.

The new text now reads "In particular, we would like to look for possible mechanisms to achieve frequency tuning (output sharply peaked at some frequency for a given input) and amplitude compression (input changes by several orders of magnitude give rise to significantly smaller changes of the output)."

P2, L9, LC: this kind of referencing is neither scientific nor informative.

We do not understand the objection. We simply provide a number of references to demonstrate that the idea of a second filter is not new and has been raised in the literature. We think it might help the reader to see arguments for such a filter.

P2, L12, LC: what is the purpose of this sentence and bunch of references?

This is a matter of writing style and philosophy. We both believe that it is beneficial, at least to some readers who want to learn more on the subject, to have quick access to useful literature.

P2, L22, LC: 'Another salient difference is that the RL is not regarded as a completely rigid body' Is the RL rigid in your model or the others? why should it be rigid?

The new text now reads "Another salient difference is that *in our models* the RL is not regarded as a completely rigid body; rather, the cuticular plates (CPs) can form mild bulges or dents in response to the local forces exerted by the corresponding OHC and OHB." The motivation for this assumption is provided for example in Section 3C; references to descriptions of the RL as a rigid beam are provided in Section 2A.

P2, L37, LC: 'but we believe that the important fact is that...' what makes you believe this?

While some models, such as those in Ref. (24,27) assume a single OHC, we found (see Section 4D) that having a second OHC enables control of the relative positions of the resonance frequencies of the BM

and of the IHB. We could add a third OHC; after all, the advantage of our platform is that we can easily change the model for any component. However, in the present work we felt that adding a third OHC would be disproportionately cumbersome in comparison to the other simplifying assumptions in the model. The text is now amended to reflect this reasoning.

P2, L40, LC: Most recent measurements contradict this. E.g., 'Ren, T., He, W., & Kemp, D. (2016). Reticular lamina and basilar membrane vibrations in living mouse cochleae. Proceedings of the National Academy of Sciences, 113(35), 9910-9915.'

We are familiar with this reference and indeed cited it in our previous version. That paper does not contradict pivot around the pillar cells head, because it does not deal with displacement as a function of x . We amended the text to clarify this point.

P2, L50, LC: [showed that the x -...]

Corrected in the new text.

P2, L51, LC: This is not correct. Ref. 20 (Fig.2) shows that x & y motions depend on the measurement location and the frequency.

The text was amended to now read: "... (23,32) showed that, for frequencies that are not too far from resonance, the amplitudes of the x - and y -component of this relative motion are of the same order of magnitude."

'which is usually disregarded' references for both assumptions are required.

The text was amended, and references were added: "... the pulsatile mode (31, 33), which is usually disregarded (27, 30). "

'Pressure exerted by endolymph on OoC components is expected' why? Your rationale or a reference?

This is one of the conclusions of our model, as stated in the present version.

Replies to Reviewer 2

Reviewer: While the author's ability to construct the model and get it to be stable is commendable, contribution to new methodology is not strong. Would like to see more detailed comparison with existing data, which is what most cochlear modelers would be interested, and this is actually the advantage of a numerical model.

Authors: The main purpose of this paper is not to provide specific results and compare to specific experiments. Rather, it is to provide a flexible instrument, in which individual models of each OoC component can be varied and the influence of this variation can be assessed. Nevertheless, several results, obtained for the particular set of models that we have considered, are compared against experimental results or against other models in sections 4A and 4B.

Fig1. The schematic drawing does not reflect the anatomical features in the OC. The Hensen cells seem to have a free end which is not true. The motion the BM is due to pressure difference between the scala media and scala tympani (ST), but I do not see ST in Fig 1.

The caption of Fig. 1 and the text in the Introduction, have been extended to clarify these points.

“There are normally three rows of outer hair cells, but we believe that the important fact is that there is more than one, and include just two outer hair cells in our explicit models.” But the BM displacement shape is complicated and two OHCs may not address this feature well.

The advantage of our platform is that we can easily change the model for any component. In particular, we can decide how many OHCs to include in an OoC slice. For instance, the models in Ref. (24,27) take just one OHC. In Section 4D we show that our model predicts that a second OHC could enable determining the relative positions of the resonance frequencies of the BM and of the IHB. We restricted ourselves to 2 OHCs, because we think that adding a third OHC would be disproportionately cumbersome in comparison to the other simplifying assumptions in the model.

Appendix B

Dear Editor,

We thank the reviewers for their numerous comments and suggestions, which have helped improve the article. We have implemented practically all of them; in the few exceptions in which a suggestion was not implemented, we explain the reason and we are confident that the reviewers will agree in light of the clarification.

Responses to reviewers

Author responses are written in bold.

Reference numbers correspond to the present version.

Line numbers differ between versions and are just approximate.

Reviewer: 2 (*The order here is the same as in the decision letter*)

Some of the assumptions made to the model are not justified properly and lack necessary supporting materials, which makes it difficult to follow the logic of the work. In the abstract and throughout the text, verb tense was used incorrectly. Some other comments can be found below:

We have added references and explanations wherever requested by the reviewers. The present version has been proofread by a professional language editor.

P1, L38~40, L: “For typical audible frequencies, the wave length of sound in the perilymph is of the same order of magnitude as the length of the entire cochlea.” This refers to the human cochlea, but there is no clear explanation that the model proposed in this paper was for the human or other species.

This sentence now reads “Across most of the frequency range audible to non-aquatic mammals (8), the wavelength of sound in the perilymph is longer than the entire cochlea;”

P1, L55, L: the BM thickness also varies along its length.

This sentence now reads “The stiffness of the BM gradually decreases by two orders of magnitude and its width increases by a factor of about four from the base to the apex of the cochlea; the longitudinal variation of the BM thickness is less conspicuous.”

P1, L52, L: “The partition is composed of the BM, the organ of Corti (OoC) and the tectorial membrane (TM)”. This description includes only three main components, but there are other important components within the cochlear partition.

By “partition” we mean an object (divider) and not an action (enumeration). To avoid confusion, we have edited this paragraph and it now reads “The partitioned structure of the cochlea is described, e.g., in (4, 6). By ‘partition’ we mean the helical strip composed of the BM, the organ of Corti (OoC), which is mounted on the BM, and the tectorial membrane (TM), located immediately above the OoC and separated from it by a thin fluid gap.”

P1, L57, L: “The BM has a large Young modulus”, the term Young modulus is usually used in the cochlear model but not for describing the physiological feature of the BM.

The present description is “The BM has highly anisotropic stiffness, mainly due to the presence of radial collagen fibres (12,13). The stiffness of the BM gradually decreases...”

P1, L11, R: The statement “pressure differences within the SM” is not proper, there is a pressure difference between the SM and ST, what does pressure difference within the SM refer to? There is another conflict with this point (P2, L25~L27, R) where the pressure in the SM will be taken equal to the pressure in the tissues under the RL and the HC.

We have edited this statement. We now write “The BM acts as an elastic barrier and is exposed to the large pressure difference between the SM and the ST, whereas the OoC and the TM are shielded from the ST.”

The considered tissues under the RL and the HC lie above the BM and the pressure in them is essentially that of the SM. Taking this pressure as uniform is an idealization of our model.

P2, L44, R: “width” should be changed to “thickness”.

We changed to “thickness”.

P2, L3, L: Is there any supporting evidence?

Do you mean P2, L3, R? To the best of our knowledge, motion of the CPs relative to the RL has not been studied thus far, and one of the objectives of this paper is the evaluation of the influence that such a motion could have. The plausibility of this possibility is discussed in Sec. 3C and examined further in Sec. 3N.

P2, L19~23, L: It would be interesting to see the difference of using three OHCs.

We would like to analyze the influence of a third OHC, as well as additional refinements of the models, but such studies are not feasible within the deadline for resubmission of this manuscript.

P2, L26, R: “the RL pivots as a rigid beam around the pillar cells head”. This was mentioned several times in the paper and was used as a reference for setting the origin. But this somewhat conflicts with the point that the authors introduced cuticular plates into the model.

The RL as a frame is taken as a rigid beam, but, in contrast to previous literature, the CPs are allowed to bend.

P2, L40~42, L: “Accordingly, except for rotational and for fluid motion, motion will be restricted to the y-direction.” The authors mentioned that the amplitudes of the x and of the y components of the relative motion are of the same order of magnitude, which implies that motions in both directions are important. (P10, L2~3, L) “According to our models, the fluid flow at the IHB region is driven by the

vertical motion of the CPs,” This is because the current setting only considers motion in the y-direction.

Indeed. What our present models study is the influence of the vertical motion of the CPs. The shearing motion has been studied by many other models.

Reviewer: 1

General comments and suggestions:

The presentation of the work has been improved significantly. Authors make a clever use of dimensionless analysis, although at the end I would suggest scaling the frequency and spatial dimensions to a specific specimen (animal or human) to provide a realistic perception of the numbers (e.g., critical frequency). The unsynchronized work generated by the OHCs are assumed to be the reason for lower vibrations, but I could not distinguish it from mechanical energy dissipation caused by the mechanical and passive damping of the system. The amplification mechanism for the slice model is thoroughly studied, but extrapolating it to the full-length cochlea needs a continuous segmentation or at least wave-dependent repetition of the slice model. For example, the amplification of the OHCs shifts the resonance to higher frequencies e.g., in fig.2M of (Lee, H. Y., Raphael, P. D., Xia, A., Kim, J., Grillet, N., Applegate, B. E., ... & Oghalai, J. S. (2016). Two-dimensional cochlear micromechanics measured in vivo demonstrate radial tuning within the mouse organ of Corti. *Journal of Neuroscience*, 36(31), 8160-8173.) the resonance peak has shifted from 5kHz at 80 dB to 10 kHz at 10 dB. It was not clear to me that model can predict the shift to make conclusion about the amplification of the traveling wave.

We thank the referee for his/her input and regret that it was not available to us in the previous revision. We have added explicit dimensional values in the first paragraph of section 4A. Passive damping of the system is present in the Navier-Stokes equation, and also in the damping coefficients in Eqs. [8], [10] and [13], and in the forces F_i . We focus on the passage of vibrations from the BM to the inner hair cell bundle, so that Fig. 5E in the paper by Lee et al. is more closely related to what we do than Fig. 2M. Only in Sec. 4D do we deal with a possible mechanism for amplification or damping of the traveling wave at the BM. The resonance frequency for amplification from the oval window to the BM does

not necessarily coincide with that of amplification from the BM to the IHB and, since both amplifications depend on the amplitude, the overall resonance frequency is expected to depend on the amplitude.

Specific comments:

Maybe it is the RSOS style, but the line numbering is still troubling.

P1,L38, RC: 'typical audible frequencies' is a spectrum from 20 to 20,000Hz, the wavelength varies by 3 order of magnitudes.

We have changed this sentence to "Across most of the frequency range audible to non-aquatic mammals (8), the wavelength of sound in the perilymph is longer than the entire cochlea;"

P1,L40, RC: First you refer to chambers as 'partitions', then in the same sentence partition is referring to a small cross-sectional region of the BM where the "energy" is "deposited". The switch between reference of the partition continues in the next sentences. The definition of 'partition' could be different as mentioned in different papers (fluid chambers partitions, cross-sectional partitions of the cochlea etc.), but should be consistent in the text.

To avoid confusion, we have edited this paragraph and 'partition' has been defined explicitly: "...such that the energy deposited on the partition is concentrated in a short segment of it [less than a millimetre (11)]. The partitioned structure of the cochlea is described, e.g., in (4, 6). By 'partition' we mean the helical strip composed of the BM, the organ of Corti (OoC), which is mounted on the BM, and the tectorial membrane (TM), located immediately above the OoC and separated from it by a thin fluid gap. The BM has highly anisotropic stiffness, mainly due to the presence of radial collagen fibres (12,13). The stiffness of the BM gradually decreases by two orders of magnitude and its width increases by a factor of about four from the base to the apex of the cochlea; the longitudinal variation of the BM thickness is less conspicuous. Regarding the partition as "horizontal," it..."

L55: 'separate collagen fibers, with length, width and stiffness that gradually vary' could you put a reference for this? I believe the collagen fibers vary in

concentration but not in width and length thorough the cochlear length (Cabezudo, L. M. (1978). The ultrastructure of the basilar membrane in the cat. Acta oto-laryngologica, 86(1-6), 160-175.)

Described in previous answer.

LC, L11: Please rewrite this sentence: 'exposed only to pressure differences within the SM, is exposed to the large pressure difference between'.

We have edited this statement. We now write "The BM acts as an elastic barrier and is exposed to the large pressure difference between the SM and the ST, whereas the OoC and the TM are shielded from the ST."

L22: 'frequency tuning (output sharply peaked at some frequency for a given input) and amplitude compression (input changes by several orders of magnitude give rise to significantly smaller changes of the output).' What are inputs and outputs? I assume the input is the mechanical excitation that is delivered to the cochlea at the oval window and the output is electrical signals of the auditory neurons. Please revise the definition of the compression.

The meanings of 'input' and 'output' were defined in P1,L21,RC. Here, the explanations in brackets are intended to remind the reader of the general meaning of 'frequency tuning' and 'compression' and not provide the specific definition used in this paper.

L28: what does 'theoretical treatments' mean?

We have changed this term to "mechanical simulations."

L29: the representation of mechanical systems with electrical circuit models is well-accepted method in mechanics of hearing. The newton law is a simple relationship between force and acceleration which analogously could be represented by Maxwell's equations in electrical systems. I am a mechanical engineer, but I don't think this is an issue to question the method.

We have deleted the remark.

L33: this is not true, there are FE models in which the cochlear amplification and nonlinearity have been implemented (e.g., Motallebzadeh, H., Soons, J. A., & Puria, S. (2018). Cochlear amplification and tuning depend on the cellular arrangement within the organ of Corti. Proceedings of the National Academy of Sciences, 115(22), 5762-5767.). Any modelling study is basically a trade of simplicity and realistic representation of a system. Even with detailed FE models, one can differentiate contribution of different components on the overall response. When you make a realistic system the interactions between the components get more complicated, but the model is still following the physical governing laws, you make a simple model, you neglect more details, and deviate from the realistic behaviour and you may not see the features that simplification cause, but what you see is easier to understand. This paragraph is not a strong argument to question other methods to advertise your method.

This paragraph now reads “... the OoC is divided into thousands of pieces, and a finite elements calculation is carried out (15, 16, 18–21). Hybrid methods have also been used [e.g. (22)]. Our approach involves postulating a simplified model for each anatomical component of the OoC, with idealised geometry and with as few elements and forces as possible, in order to capture the features that are essential for its functioning. After the models are chosen, Newton’s laws can be meticulously followed. Clearly, by following this approach we depart from reality, but we gain a simple and transparent way of relating between qualitative assumptions in the model and their influence on the OoC behaviour. An appealing feature of our approach is that nonlinearity, including the possibility of bifurcations, can be incorporated naturally from the assumed physiology of the OHC and OHB, without invoking parameters that depend on the sound pressure level [e.g. gain factor (20,22)].”

L43: ‘Substantial evidence(s) has(have) led ...

We now write “A well-established conclusion is...”

L57: In recent models (full length cochlear models) the pressure is a function of place and time as well (e.g., Sasmal, A., & Grosh, K. (2019). Unified cochlear model for low-and high-frequency mammalian hearing. Proceedings of the National Academy of Sciences, 116(28), 13983-13988. Motallebzadeh, H., Soons, J. A., & Puria, S. (2018). Cochlear amplification and tuning depend on the cellular arrangement within the organ of Corti. Proceedings of the National Academy of Sciences, 115(22), 5762-5767.). Do you refer any specific work that considers the RL as a rigid body? I can think of few paper (e.g., Lim, K. M., & Steele, C. R. (2002). A three-dimensional nonlinear active cochlear model analyzed by the WKB-numeric method. Hearing research, 170(1-2), 190-205.) in which this assumption is made to find low-frequency behaviour or force transmissions within the OoC.

The paper by Sasmal et al. (2019) considers the pressure in the SV+SM and in the ST, not in the subtektorial channel. The paper by Motallebzadeh et al. (2018) does not incorporate the subtektorial channel. We have modified this paragraph and it now reads “A salient departure of our models from the bulk of the literature on the subject is that we do not envision the RL as a featureless body; rather, the cuticular... exerted by the endolymph and by the corresponding OHC and OHB. A related attribute is that we take into account the local pressures in the subtektorial channel relative to that in the SM.” References to the claim that the RL behaves as a rigid beam, based on experiments, are given in P2,L28,RC.

P3, L14, RC: Do you have any specific reference for this fact that the R number is very small?

After Eq. [2] we now write: “Estimating the order of magnitude of the fluid velocity as being similar to that of the BM velocity, which even for 100 dB SPL does not exceed 10^{-2} m/s (2), leads to a Reynolds number that is at most of the order of 10^{-1} , so that the flow is certainly laminar. Invoking incompressibility, using the units defined in section 2B, and noting that the term quadratic in velocity is smaller by at least an order of magnitude than the linear terms, the x-component of the Navier–Stokes momentum equation reduces to”

‘rotational equation of motion’ -- > ‘momentum equation’ (motion equations refer to kinematics and momentum equations refer to the dynamics or force

balance)

'equation of motion'  'momentum equation'

These terms are used differently in physics and in fluid mechanics. In Eq. [5] and similar, we apply Newton's second law (linear or rotational) to a rigid body (not to a fluid volume). Since RSOS is an interdisciplinary journal, we believe that the usage within the author's discipline should be preferred.

P3, L33: what is Greek letter ' ν ' in this term? Is it velocity, or volume velocity or something else? Is it the same letter in Equation 1 (there are two Greek letters similar to ν)? Is yes, please be consistent.

The letter 'nu' (ν), defined in P3,L32,LC, is the kinematic viscosity. At P3,L57,LC we now write "... by $v(x, y, t)$ (not to be confused with the kinematic viscosity ν) the x -component of the local velocity."

P4, L37, RC: There are more recent measurements on the TM mechanical properties (e.g., Sellon, J. B., Ghaffari, R., & Freeman, D. M. (2018). The tectorial membrane: mechanical properties and functions. Cold Spring Harbor perspectives in medicine, a033514.)

At P4, L42, LC we write "Its static Young modulus...", and at P4, L45, LC we add "The poroelastic, electrokinetic, longitudinal-radial coupling, and wave properties of the TM are reviewed in (46)."

Equation 7: Please define 'sgn' function

We have added the definition.

P4, L52: please elaborate 'work performed by the bundle motility vanishes'. Work is performed by force do you mean that hair bundles, in addition to the OHC body, generate force?

Yes, this is the central finding of Ref. (47).

Line 27: Please elaborate on 'non vanishing work'. Damped or dissipated energy refers to energy loss due to the damping or friction but here I believe you are referring to the work due to the OHC motility.

Yes, we now write "... acts, OHC motility *can*..."

P5, L11: remind the reader that the length 1 is in the adopted normalized coordinate system.

We now write "... almost 1, i.e., it almost touches the TM."

L46: please elaborate on 'we do not consider noise that arises in the OoC itself'. What kind of noise the OoC can produce and what exactly noise mean here? Do you mean perturbations due to the nonuniform distribution of the mechanical properties?

We now write "we do not consider noise due to thermal fluctuations in the OoC itself."

'A. Main Results' is not a specific subtitle to differentiate from the rest of the results. Maybe 'summary of key findings'?

We now write "Key findings"

L 37, LC: please elaborate on 'self-oscillations (non zero output for zero input)'. Do you mean the system is unstable or underdamped in this specific situation?

We now write "...the OoC is unstable and undergoes self-oscillations (non-zero output for zero input), whereas for $\Delta < \Delta_c$ it is stable."

P5, L42, RC: Could you translate the units and dimensions for a specific specimen so the reader can have an idea of the critical frequency? 'corresponds to a contraction of a few percent' of a dimensionless parameter is not enough to evaluate its value.

At the end of this paragraph we write “For $\nu = 7 \times 10^{-7} \text{ m}^2 / \text{ s}$ and $D_0 = 5 \times 10^{-6} \text{ m}$, the critical frequency is $\omega_c / 2\pi = 24 \text{ kHz}$ and the critical contraction is $\Delta_c = 1.3 \mu\text{m}$. The length of an OHC is typically $\sim 10D_0$, so that its critical contraction corresponds to a few percent of its length.”

L52, RC: ‘then the OoC would provide additional tuning; if it is not, the OoC would provide an alternative mechanism for tuning’ what is the difference of ‘additional tuning’ and ‘alternative mechanism for tuning’?

We now write “In this case, if ω_c is near the frequency of the sound wave that is picked by the BM at the considered slice position, then the OoC provides additional tuning, i.e., vibrations of the IHB are more sharply tuned than those of the BM; if ω_c is far from the BM resonance frequency, the OoC provides an alternative mechanism for tuning, i.e., IHB vibrations can be tuned at a frequency at which the BM does not resonate. If ω_c is moderately close to the BM resonance frequency, the OoC can provide moderate additional tuning and shift the resonance frequency of the IHB from that of the BM. In the case that Δ is below, but moderately close to Δ_c , then the second filter can...”

P8, L53 LC: ‘is in the range between 0° and 180° (or equivalent)’ please clarify. What is equivalent to 0-180? It could be either pi plus/minus or 2pi plus/minus, depending on the periodicity function.

We now write “ [or equivalent, i.e., larger than $2n \times 180^\circ$ and smaller than $(2n + 1) \times 180^\circ$ for some integer n]”

P8, L41 RC: ‘Within a more realistic model, the energy exchange described here should be regarded as a contribution.’ Contribution to?

We now write “... contribution to the energy exchange between the BM and the OoC.”

P8, L43 RC: The attenuation according to the calculations here, is coming from

unsynchronized OHC force, but is there any direct calculation of the attenuations due to the damping of the system? Is it possible even when OHC provides constructive force (and work) that work is dissipated due to the mechanical damping?

Dissipations within the OoC are taken into account through viscosity and damping coefficients, but dissipation by the BM is beyond the scope of this study.

P10, 5, LC: although HC's are outside the subtektorial space, but do to their large deformations (because of lower stiffness), could they also pump fluid into the subtektorial space?

Yes, we have added this at P10,L44,LC: "An intuitive explanation could be that due to their lower stiffness (neglected in our models) the HC undergo comparatively large deformations and can pump fluid into and out of the subtektorial space."

Appendix C

Dear Editor,

Thank you for your mail of August 9.

We have revised our manuscript as specified below.

We thank Reviewer 1 for his/her detailed remarks.

Sincerely,

J. Berger and J. Rubinstein

P1,4, RC: “The BM acts as an elastic barrier and is exposed to the large pressure difference between the SM and the ST”, the pressure gradient between SM and ST varies along the length as a function of frequency. Apical to the characteristic frequency there is almost no pressure difference between these two chambers.

We now write: “...exposed to the pressure difference between the SM and the ST, which is large where vibrations are present”

P2 26-46, LC: In my opinion the main feature of this work in implementing analytical approach to solve the mechanics of the OoC. FE studies use numerical discretion methods and lumped element models use electrical analogy.

We have added: “In this way, most of the analysis can be kept analytical, and only a slight numerical treatment is required.”

P2 2-4 RC: RL and CPs are making a continues frame, which is flexible at CPs and rigid at RL segments. Authors emphasize on the RL characteristics ‘A salient’ feature of their method and a significant deviation from ‘featureless body’ from the whole studies in the literature. There are studies of both rigid and flexible RL and I suggest to emphasize on more important characteristics on the study.

We have replaced this statement with “Our models depart from the bulk of the literature on the subject by considering the cuticular plates (CPs) and the RL as separate bodies; the CPs can form...”

P3 14-15 RC: “fluid velocity as being similar to that of the BM velocity”. Due to the no-slip boundary condition of the NS equations on the BM and fluid interface, their velocities are exactly the same (not similar) at the BM surface.

We now write: “...fluid velocity in the subtektorial channel as being similar to that of the fluid in contact with the BM, i.e., the BM...”

P5 51, RC: Is this instability part of the OoC physiology which results in spontaneous otoacoustic emissions?

At the end of the paragraph in P6 31 LC we have added: “As explained in (55), fluctuations that lead to $\Delta > \Delta_c$ result in spontaneous otoacoustic emissions with frequency $\omega_c / 2\pi$.”

P6 19 LC: The hearing frequency is more common than angular frequency (ω) in hearing acoustics, although they are linearly related, but I would suggest converting them to actual frequency (as done in line 12 the same page).

We have replaced ω_c with $\omega_c / 2\pi$ in this paragraph.

P9 F. Nonlinearity: I could not follow the argument of nonlinearity due to the non-harmonic θ . What kind of nonlinearity is referred here? How does non-sinusoidal θ and the Fourier transfer of it establish nonlinearity?

We now write: “Since the constitutive relations [7], [9] and [14] are nonlinear, it is not a surprise that a sinusoidal input can result in a non-sinusoidal output, but rather contain higher harmonics. For example, if the peak to peak amplitude of the BM vibration is $0.002D_0$ and its frequency is $\omega_R / 2\pi$, then the vibration of the IHB (after the periodic regime is established) is not sinusoidal, but is rather approximately proportional to

$\cos \omega_R t + 0.034 \cos[3(\omega_R t + 0.70)] + 0.005 \cos[5(\omega_R t - 0.34)]$. **Expanding θ_{in} in a**

Fourier series, $\theta_{in} = \sum_{n=0}^{\infty} a_n \cos[n(\omega_{BM} t + \phi_n)]$, we studied the amplitude

dependence of the coefficients at the resonant frequencies. We obtained that the even..."